# Bandwidth-based Step-Sizes for Non-Convex Stochastic Optimization: Non-monotonicity, Worst-case Convergence and Performance

## Abstract

Many popular learning-rate schedules for deep neural networks combine a decaying trend with local perturbations that attempt to escape saddle points and bad local minima. We derive convergence guarantees for bandwidth-based step-sizes, a general class of learning-rates that are allowed to vary in a banded region. This framework includes many popular cyclic and non-monotonic step-sizes for which no theoretical guarantees were previously known. We provide worst-case guarantees for SGD on smooth non-convex problems under several bandwidth-based step sizes, including stagewise $1/\sqrt{t}$ and the popular *step-decay* ("constant and then drop by a constant"), which is also shown to be optimal. Moreover, we show that its momentum variant converges as fast as SGD with the bandwidth-based step-decay step-size. Finally, we propose novel step-size schemes in the bandwidth-based family and verify their efficiency on several deep neural network training tasks.

## 1 Introduction

Stochastic gradient methods including stochastic gradient descent (SGD) (Robbins and Monro, 1951) and its accelerated variants (e.g., SGD with momentum (Polyak, 1964; Sutskever et al., 2013)) have become the algorithmic workhorse in much of machine learning. The step-size (learning rate) is the most important hyper-parameter for controlling the speed at which gradient-based methods converge to stationarity. For problems with multiple local minima, the step-size also affects which local optimum the optimization process converges to. It therefore needs to be both well-designed and well-tuned to make SGD and its variants effective in practice.

In the deep learning literature, cyclical step-sizes (Loshchilov and Hutter, 2017; Smith, 2017) and non-monotonic schedules (Keskar and Saon, 2015; An et al., 2017; Seong et al., 2018; Loizou et al., 2021) have attracted strong recent interest, with significant benefits for non-convex problems with poor local minima or saddle points (Seong et al., 2018). Popular cyclical schedules include the cosine step-size (cosine with restart) (Loshchilov and Hutter, 2017) and the triangular policy (Smith, 2017), which have become the default choices in some deep learning libraries, e.g., PyTorch and TensorFlow (cf. lr_scheduler.CyclicLR and CosineAnnealingLR). However, non-monotonic policies are much more complex to analyze than decaying ones, and theoretical results for these non-monotonic policies are scarce. This motivates us to focus on a bandwidth step-size framework, in which

$$m\delta(t) \leq \eta^t \leq M\delta(t) \tag{1}$$

for some boundary function $\delta(t)$ and positive constants $m$ and $M$. This framework allows for non-monotonic step-sizes and covers most of the situations discussed above. In particular, it includes the cosine (Loshchilov and Hutter, 2017), triangular (Smith, 2017), sine wave (An et al., 2017) step-sizes as special cases. The framework provides a uniform convergence rate guarantee for all step-size policies which remain in the band (1). This gives a lot of freedom to design novel step-sizes schedules with improved practical performance without loosing track of their theoretical convergence guarantee.

The generic bandwidth framework has recently been proposed by Wang and Yuan (2021), but they only analyzed strongly convex problems. We believe that more significant potential lies in the non-convex regime. For non-convex problems, non-monotonic step-sizes have distinct advantages, helping iterates to escape local minima and producing final iterates of high quality. In the paper, we

demonstrate this point on both a simple toy example and on large-scale neural network training tasks. Our main contribution is a sequence of non-asymptotic convergence results for the bandwidth step-size on non-convex optimization problems, based on the popular "constant and then drop" step-size schedules (Krizhevsky et al., 2012; He et al., 2016; Hazan and Kale, 2014; Ge et al., 2019; Wang et al., 2021). This allows non-monotonic variations both within each (inner) stage and between stages.

## 1.1 CONTRIBUTIONS

Inspired by the strong potential of non-monotonic step-size schedules demonstrated above, we extend the bandwidth-based step-size framework to *"constant and then drop"* (multi-stage) profiles, where the bands stay constant throughout each stage and drops between stages. We provide convergence guarantees for both SGD and its momentum variant (SGDM) on non-convex problems. Specifically,

- We establish worst-case theoretical guarantees for SGD with bandwidth step-size on smooth nonconvex problems. We **(i)** derive an optimal rate for SGD under a bandwidth step-size with $\delta(t) = 1/\sqrt{t}$; **(ii)** and achieve optimal and near-optimal rates for step-decay (constant and then drop by a constant), improving the results by Wang et al. (2021).

- We provide worst-case theoretical guarantees for SGDM with bandwidth-based step-decay step-size in the smooth non-convex setting. To the best of our knowledge, these are the first results that provide optimal (Theorem 4.3) and near-optimal (Theorem 4.2) results for momentum with *step-decay* step-sizes. Moreover, our results significantly improve the convergence results from Liu et al. (2020) (see Remark 4.4).

- Our analysis results also provide state-of-the-art theoretical guarantees for *cosine* (Loshchilov and Hutter, 2017) and *triangular* (Smith, 2017) step-sizes if their boundary functions are within our bands. Especially, we improve the result of Li et al. (2021) for cosine step-size and achieve a state-of-art rate (see Remark 3.4). Moreover, our results first provide the convergence guarantees for triangular step-size (Smith, 2017).

- We propose novel, possibly non-monotonic, step-size schedules (e.g., step-decay with linear-mode and cosine-mode) based on the bandwidth-based framework and demonstrate their efficacy on several large-scale neural network training tasks.

## 1.2 RELATED WORK

This subsection reviews the theoretical development of the SGD algorithm and its momentum variant in the smooth non-convex setting, with a special focus on different step-size policies.

**SGD for nonconvex problems** The first non-asymptotic convergence of SGD to a stationary point of a general smooth non-convex function was established in Ghadimi and Lan (2013). The authors proved that a constant step-size $\mathcal{O}(1/\sqrt{T})$ attains a convergence rate of $\mathcal{O}(1/\sqrt{T})$, where $T$ is the iteration budget. To the best of our knowledge, this rate is not improvable and was proven to be tight up to a constant without additional assumptions (Drori and Shamir, 2020). For the $1/\sqrt{t}$ decay step-size, an $\mathcal{O}(\ln T/\sqrt{T})$ rate can be easily obtained from (Ghadimi and Lan, 2013). This rate can be improved to the optimal by selecting a random iterate using weights proportional to the inverse of the step-size (Wang et al., 2021). The sampling rule in (Wang et al., 2021) depends on the step-size and is easily applicable to different step-size policies. Thus, in this paper, we choose a similar sampling rule as (Wang et al., 2021) to favor the later iterates when selecting the output for SGD and its momentum variant.

**Step-decay step-sizes** Recently, the theoretical performance of step-decay or stagewise strategies has attracted an increasing attention due to their excellent practical performance (Yuan et al., 2019; Ge et al., 2019; Chen et al., 2019; Li et al., 2021; Wang et al., 2021). For a class of least-squares problems, Ge et al. (2019) established a near-optimal $\mathcal{O}(\ln T/T)$ rate for the step-decay step-size (cut by 2 every $T/\log_2(T)$ iterates) and showed that step-decay can perform better than the polynomial decay step-size. Stochastic optimization methods with stagewise step-sizes decaying as $1/t$ were analyzed in Chen et al. (2019). A near-optimal rate for the continuous version of step-decay, called exp-decay, as well as for cosine decay step-sizes under the Polyak-Lójasiewicz (PL) condition and a general smooth assumption were established in Li et al. (2021). However, in the smooth case, to achieve such results for exponential and cosine decay step-sizes, the initial step-size is required to

be bounded by $\mathcal{O}(1/\sqrt{T})$. This is obviously impractical when the number of iterations $T$ is large. Near-optimal rates (up to $\ln T$) of SGD with step-decay step-size in several general settings including strongly convex, convex and smooth (non-convex) problems were proved in Wang et al. (2021). They also removed the restriction on the initial step-size for exponential decay step-sizes. Empirical evidences have been given in (Wang and Yuan, 2021) that bandwidth-based strategies can improve the performance of the step-decay step-size on some large scale neural network tasks. However, no theoretical guarantees for non-convex problems were given.

**SGD with momentum on nonconvex problems** The momentum variant of SGD (SGDM) has been widely used in deep neural networks (Krizhevsky et al., 2012; Sutskever et al., 2013; He et al., 2016; Zagoruyko and Komodakis, 2016). Due to its practical success on neural networks, its theoretical performance is now attracting a lot of interest, especially for nonconvex problems (Yan et al., 2018; Gadat et al., 2018; Chen et al., 2019; Gitman et al., 2019; Mai and Johansson, 2020; Liu et al., 2020; Defazio, 2020). Under the assumption of bounded gradients, Yan et al. (2018) proposed a unified analysis framework for stochastic momentum methods and proved an optimal $\mathcal{O}(1/\sqrt{T})$ rate under constant step-sizes. A similar result for the Nesterov-accelerated variant was established in Ghadimi and Lan (2016). However, studies related to the multi-stage performance of SGD with momentum is lacking and far from being complete. Reference Chen et al. (2019) considers a momentum method with a stagewise step-size, but the method is a proximal point algorithm with extra averaging between stages, and not the widely used momentum SGD considered here. More recently, Liu et al. (2020) established the convergence for multi-stage SGDM and provided empirical evidence to show that multi-stage SGDM is faster. However, their results require an inverse relationship between stage length and step-size which limits the initial stage length or step-size. A detailed comparison with (Liu et al., 2020) will be given in Section 4 (see Remark 4.4).

**Organization:** The rest of this paper is organized as follows. Notations and basic definitions are introduced in Section 2. Our novel theoretical results for SGD and its momentum variant (SGDM) under bandwidth-based step-sizes are introduced in Sections 3 and 4, respectively. Numerical experiments are presented and reported in Section 5. Finally, Section 6 concludes the paper.

## 2 PROBLEM SET-UP

We study the following, possibly non-convex, stochastic optimization problem

$$\min_{x \in \mathbb{R}^d} f(x) = \mathbb{E}_{\xi \sim \Xi}[f(x;\xi)] \tag{2}$$

where $\xi$ is a random variable drawn from some (unknown) probability distribution $\Xi$ and $f(x;\xi)$ is the instantaneous loss function over the variable $x \in \mathbb{R}^d$. We consider stochastic gradient methods that generate iterates $x^t$ according to

$$x^{t+1} = x^t - \eta^t d^t \tag{3}$$

where $\eta^t$ is the step-size and $d^t$ the search direction (e.g., $d^t = \nabla f(x^t;\xi)$ for SGD). We assume that there are constants $m > 0$ and $M \geq m$, and two functions $n(t)$ and $\delta(t)$: $\mathbb{R} \to \mathbb{R}$ such that such that

$$\eta^t = n(t)\delta(t), \ \forall\, t \geq 1,$$

where $n(t) \in [m, M]$ and $\delta(t)$ is monotonically decreasing function satisfying $\delta(1) = 1$. Note that even though the boundary function $\delta(t)$ is monotonic, the step-size itself is not restricted to be. Throughout the paper, we make the following assumptions:

**Assumption 1.** *The loss function $f$ satisfies $\|\nabla f(x) - \nabla f(y)\| \leq L\|x - y\|$ for every $x, y \in dom\,(f)$.*

**Assumption 2.** *For any input vector $x$, the stochastic gradient oracle $\mathcal{O}$ returns a vector $g$ such that (a) $\mathbb{E}[\|g - \nabla f(x)\|^2] \leq \rho \|\nabla f(x)\|^2 + \sigma$ where $\rho \geq 0$ and $\sigma \geq 0$; (b) $\mathbb{E}[\|g\|^2] \leq G^2$.*

## 3 NON-ASYMPTOTIC CONVERGENCE OF SGD WITH BANDWIDTH-BASED STEP-SIZE

In this section, we provide the first non-asymptotic convergence guarantees for SGD with bandwidth-based step-sizes on smooth non-convex problems. The results consider a general family of bandwidth-based step-sizes which includes the classical multi-stage SGD as a special case.

---

**Algorithm 1** SGD with Bandwidth-based Step-Size

---

1: **Input:** initial point $x_1^1$, # iterations $T$, # stages $N$, stage length $\{S_t\}_{t=1}^N$ such that $\sum_{t=1}^N S_t = T$, the sequences $\{\delta(t)\}_{t=1}^N$ and $\left\{\{n(t,i)\}_{i=1}^{S_t}\right\}_{t=1}^N \in [m, M]$ with $0 < m \leq M$

2: **for** $t = 1 : N$ **do**

3:    **for** $i = 1 : S_t$ **do**

4:       Query a stochastic gradient oracle $\mathcal{O}$ at $x_i^t$ to get a vector $g_i^t$ such that $\mathbb{E}[g_i^t \mid \mathcal{F}_i^t] = \nabla f(x_i^t)$[1]

5:       Update step-size $\eta_i^t = n(t,i)\delta(t)$

6:       $x_{i+1}^t = x_i^t - \eta_i^t g_i^t$

7:    **end for**

8:    $x_1^{t+1} = x_{S_t+1}^t$

9: **end for**

10: **Return:** $\hat{x}_T$ is uniformly chosen from $\left\{ x_1^{t^*}, x_2^{t^*}, \cdots, x_{S_{t^*}}^{t^*} \right\}$, where the integer $t^*$ is chosen from $\{1, 2, \cdots, N\}$ with probability $P_t = \delta^{-1}(t)/(\sum_{l=1}^N \delta^{-1}(l))$

---

Algorithm 1 details our bandwidth-based version of the popular "constant and then drop" policy for SGD. Here, the boundary function $\delta(t)$ is adjusted in an outer stage, and the length of each stage $S_t$ is allowed to vary. Similar to Wang et al. (2021), the output distribution depends on the inverse of $\delta(t)$, hence puts more weight on the final iterates. By considering specific combinations of $\delta(t)$ and $S_t$, this framework allows us to analyze several important multi-stage SGD algorithms, including those with constant, polynomial-decay and step-decay step-sizes. For example, we consider the step-decay step-size by letting $n(t,i) = m$ and $\delta(t) = 1/\alpha^{t-1}$ where $m$ denotes its initial step-size and $\alpha > 1$. Many interesting results on polynomial-decay step-size (e.g., $\delta(t) = 1/\sqrt{t}$, we called it $1/\sqrt{t}$-band) are given in Appendix A.

### 3.1 Convergence Under Bandwidth Step-Decay Step-Size

Another important step-size is Step-Decay ("constant and then drop by a constant"), which is popular and widely used in practice, e.g. for neural network training (Krizhevsky et al., 2012; He et al., 2016). In this subsection, we analyze bandwidth step-sizes that include step-decay as a special case.

For Step-Decay, the stage length $S_t$ is typically a hyper-parameter selected by experience. We first analyze a bandwidth version of the algorithm analyzed in [Theorem 3.2](Wang et al., 2021), namely Algorithm 1 with $N = \lfloor (\log_\alpha T)/2 \rfloor$ outer loops where $\alpha > 1$, each with a constant length of $S_t = \lceil 2T/\log_\alpha T \rceil$. The logarithmic dependence of $N$ on $T$ leads to a small number of stages in practice, and was demonstrated to perform well in deep neural network tasks (Wang et al., 2021).

**Theorem 3.1.** *Under Assumptions 1 and 2(a), and assume that there exists a constant $\Delta_0 > 0$ such that $\mathbb{E}[f(x_1^t) - f^*] \leq \Delta_0$ for each $t \geq 1$ where $f^* = \min f(x)$, if we run Algorithm 1 with $T > \alpha^2$, $\eta_i^t \leq 1/((\rho + 1)L)$, $N = \lfloor (\log_\alpha T)/2 \rfloor$, $S_t = \lceil 2T/\log_\alpha T \rceil$, and $\delta(t) = 1/\alpha^{t-1}$ for $1 \leq t \leq N$, where $\alpha > 1$ then*

$$\mathbb{E}[\|\nabla f(\hat{x}_T)\|^2] \leq \left( \frac{\Delta_0}{2\alpha m} + \frac{\alpha M^2 L \sigma}{2m} \right) \frac{(\alpha - 1)}{\ln \alpha} \cdot \frac{\ln T}{\sqrt{T} - \alpha}.$$

Theorem 3.1 establishes a near-optimal (up to $\ln T$) rate for the step-decay bandwidth scheme which matches the result achieved at its boundaries i.e., $\eta_i^t = m\delta(t)$ or $\eta_i^t = M\delta(t)$ (Wang et al., 2021). As the next theorem shows, this guarantee can be improved by appropriate tuning of the stage length $S_t$.

**Remark 3.2.** *(**Justification of uniformly bounds on the function values**) In Theorem 3.1, we require that the expectation of the function value at each outer iterate $\mathbb{E}[f(x_1^t)]$ is uniformly upper bounded. As shown by Shi et al. (2020), the function values at the iterates of SGD can be controlled (bounded) by the initial state provided the step-size is bounded by $1/L$. So the assumption is fair if the initial state is settled. Nevertheless, this assumption (or its stronger version that the objective function is bounded) is commonly used or implied in optimization (Hazan et al., 2015; Xu et al., 2019b; 2020; 2019a) and statistic machine learning (Vapnik, 1998; Cortes et al., 2019) , and it has never been violated in our numerical experiments.*

---

[1] We use $\mathcal{F}_i^t$ to denote $\sigma$-algebra formed by all the random information before current iterate $x_i^t$ and $x_i^t \in \mathcal{F}_i^t$.

**Theorem 3.3.** *Under Assumptions 1 and 2(a), and assume that there exists a constant $\Delta_0 > 0$ such that $\mathbb{E}[f(x_1^t) - f^*] \leq \Delta_0$ for each $t \geq 1$, if we run Algorithm 1 with $T > \alpha^2$, $\eta_i^t \leq 1/((\rho + 1)L)$, $S_0 = \sqrt{T}$, $S_t = \lceil S_0 \alpha^{(t-1)} \rceil$ and $\delta(t) = 1/\alpha^{t-1}$ where $\alpha > 1$, then*

$$\mathbb{E}[\|\nabla f(\hat{x}_T)\|^2] \leq \frac{\alpha + 1}{\alpha - 1} \left( \frac{2\Delta_0}{m} + \frac{M^2 L\sigma}{m} \right) \frac{1}{\sqrt{T}} + \mathcal{O}\left( \frac{1}{T} \right).$$

**Optimal rate for step-decay step-size** The theorem shows that if the stage length $S_t$ increases exponentially, and the length of the first stage is set appropriately, then we can achieve an optimal $\mathcal{O}(1/\sqrt{T})$ rate for the bandwidth step-decay step-size in the non-convex case. If $M = m$, which means that the bandwidth scheme degenerates to the step-decay type step-size, Theorem 3.3 removes the logarithmic term present in the results of Wang et al. (2021). To the best of our knowledge, this is the first result that demonstrates that vanilla SGD with step-decay step-sizes can achieve the optimal rate for general non-convex problems. The numerical performance of the two step-size schedules in Theorems 3.1 and 3.3 are reported in Figure 4.

**Benefits of Theorems 3.5 vs the references of Hazan and Kale (2014); Yuan et al. (2019)** Another commonly used step-decay scheme in theory which halves the step-size after each stage and then doubles the length of each stage (e.g., (Hazan and Kale, 2014; Yuan et al., 2019)). In Hazan and Kale (2014), which considers strongly convex problem, the initial stage is very short, $S_1 = 4$, while the analysis in Yuan et al. (2019) for PL functions use an inverse relation between stage length and step-size, which means that a longer initial stage length requires a smaller stepsize. In contrast to these references, Theorem 3.3 considers a step-decay with a long first stage, $S_1 = \lceil \sqrt{T} \rceil$, which allows us to benefit from a large constant step-size for more iterations.

**Remark 3.4.** *(**Guarantees for cyclical step-sizes**) In (Loshchilov and Hutter, 2017), the authors decay the step-size with cosine annealing and use $\eta_{\min}^t < \eta_{\max}^t$ to control the range of the step-size. If $m\delta(t) \leq \eta_{\min}^t, \eta_{\max}^t \leq M\delta(t)$, then our results provide convergence guarantees for their step-size. To achieve a near-optimal rate, Li et al. (2021, Theorem 4) need to use an initial step-size that is smaller than $\mathcal{O}(1/\sqrt{T})$ which is obviously impractical. In contrast, we allow the cosine step-size to start from a relatively large step-size and then gradually decay (see Theorem 3.1) and also improve the convergence rate to be optimal (Theorem 3.3).*

*A triangular cyclical step-size is proposed by Smith (2017) which is varied around the two boundaries that drop by a constant after a few iterations. Our analysis first provides theoretical guarantees (e.g., Theorems 3.1 and 3.3) also for this step-size. The details are shown in Appendix D.2.*

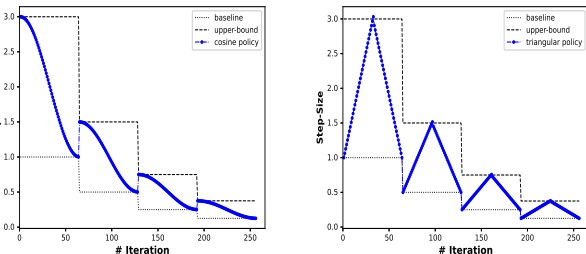

Figure 1: Cosine step-size (left) and triangular step-size (right)

The bandwidth step-sizes we consider above are independent on the random information. In recent years, some non-monotonic step-sizes have been proposed that are dependent on the current random information, e.g., the trust-region-ish algorithm (Curtis et al., 2019) and stochastic Polayk step-size (Loizou et al., 2021). We provide some interesting results for these step-sizes (see Lemma B.2 in Appendix B).

# 4 NON-ASYMPTOTIC CONVERGENCE OF SGDM UNDER BANDWIDTH-BASED STEP-SIZE

In this section, we establish the first non-asymptotic convergence properties of SGD with momentum (SGDM) under the bandwidth-based step-size on smooth nonconvex problems.

In this scheme, the inner iterations in Step 6 of Algorithm 1 are essentially replaced by

$$v_{i+1}^t = \beta v_i^t + (1 - \beta)g_i^t \tag{4}$$

$$x_{i+1}^t = x_i^t - \eta_i^t v_{i+1}^t \tag{5}$$

for $\beta \in (0, 1)$. We refer to Algorithm 2 in Appendix C for a more detailed description.

As in many studies of momentum-methods (e.g. Ghadimi et al. (2015); Yan et al. (2018); Liu et al. (2020); Mai and Johansson (2020)), we establish an iterate relationship of the form $\mathbb{E}[W^{t+1}] \leq \mathbb{E}[W^t] - c_0\eta\mathbb{E}[e^t] + c_1\eta^2$, where $W^t$ is a Lyapunov function, $e^t$ is a performance measure (here, $e^t = \|\nabla f(\cdot)\|^2$), $\eta$ is the step-size and $c_0$ and $c_1$ are constants. However, due to the time-dependent and possibly non-monotonic bandwidth-based step-size, we cannot use the Lyapunov functions suggested in Yan et al. (2018); Liu et al. (2020) but rely on the following non-trivial construction:

**Lemma 4.1.** *Suppose that Assumption 1 and Assumption 2(b) hold. Let $z_i^t = (1-\beta)^{-1}(x_i^t - \beta x_{i-1}^t)$ and assume that there exists a constant $\Delta_0$ such that $\mathbb{E}[f(x_i^t) - f^*] \leq \Delta_0$ for $t, i \geq 1$ and the step-size in each stage is monotonically decreasing. Define the function $W_{i+1}^t$*

$$W_{i+1}^t = \frac{f(z_{i+1}^t) - f^*}{\eta_i^t} + \frac{r\left\|x_{i+1}^t - x_i^t\right\|^2}{\eta_i^t} + 2r[f(x_{i+1}^t) - f^*],$$

*where $r = \frac{\beta L}{2(1-\beta^2)(1-\beta)^2}$. Then, if $\eta_i^t \leq 1/L$, for any $t$ and $i \geq 2$, we have*

$$\mathbb{E}[W_{i+1}^t \mid \mathcal{F}_i^t] \leq W_i^t + A_1\left(\frac{1}{\eta_i^t} - \frac{1}{\eta_{i-1}^t}\right) - \left\|\nabla f(x_i^t)\right\|^2 + \eta_i^t \cdot B_1 G^2. \tag{6}$$

*where $A_1 = \frac{\beta\Delta_0}{1-\beta} + \Delta_z + \frac{rG^2}{L^2}$, $B_1 = r(1-\beta)(2-\beta) + \frac{L}{2(1-\beta)^2}$, and $\Delta_z = \frac{\Delta_0}{1-\beta} + \frac{\beta G^2}{2(1-\beta)^2 L}$.*

Note that even though the step-size is assumed to be monotonically decreasing in each stage, it may be increased between stages, leading to a globally non-monotonic step-size. The proposed bandwidth-based step-sizes (e.g., step-decay with linear or cosine modes) in the numerical experiments and the cosine annealing policy proposed in (Loshchilov and Hutter, 2017) all satisfy this condition. Note that, unlike (Mai and Johansson, 2020; Liu et al., 2020), the momentum parameter $\beta$ does not rely on the step-size, but can be chosen freely in the interval $(0, 1)$. In particular, our analysis supports the common choice of $\beta = 0.9$ used as default in many deep learning libraries (Krizhevsky et al., 2012; He et al., 2016). Similar to Remark 3.2, the function value of the iterates for momentum can also be controlled (bounded) by the initial state given $\eta_i^t \leq 1/L$; see (Shi, 2021). Therefore, we believe our assumptions are reasonable.

If we restrict the analysis to a single stage, $N = 1$, the lemma allows to recover the optimal rate for SGDM under the step-size $\eta_i^t = \eta_0/\sqrt{T}$ (Yan et al., 2018; Mai and Johansson, 2020; Liu et al., 2020; Defazio, 2020) and to prove, for the first time, an optimal $\mathcal{O}(1/\sqrt{T})$ rate for SGDM under the $1/\sqrt{i}$ stepsize. These results are formalized in Appendix D.1.

## 4.1 CONVERGENCE OF SGDM FOR BANDWIDTH STEP-DECAY STEP-SIZE

We now show the convergence complexity of SGDM with the bandwidth step-decay step-size. Here *step-decay* means that the bandwidth limits are divided by a constant after some iterations.

We first consider the total number of iterations $T$ to be given, the stage length $S_t$ to be constant, and the number of stages $N$ as a hyper-parameter.

**Theorem 4.2.** *Assume the same setting as Lemma 4.1. If given the total number of iterations $T \geq 1$, $N \geq 1$, $S_t = S = \lceil T/N \rceil$, $\delta(t) = 1/\alpha^{t-1}$ for each $1 \leq t \leq N$ and $\alpha > 1$, then*

$$\mathbb{E}[\|\nabla f(\hat{x}_T)\|^2] \leq \frac{W_1^1 \cdot N}{T\alpha^{N-1}} + (\alpha C_0 + C_2) \cdot \frac{N}{T} + \frac{(\Delta_z + C_1)}{m} \cdot \frac{N\alpha^N}{T} + MB_1G^2 \cdot \frac{N}{\alpha^{N-1}} \tag{7}$$

*where $C_0 = r(\frac{G^2}{L} + 2\Delta_0)$, $C_1 = A_1 + \Delta_z + \frac{\Delta_0}{1-\beta}$, and $C_2 = C_0 + A_2G^2$, $A_2 = 1 + \frac{\beta}{2(1-\beta)^2}$, and $W_1^1$, $A_1$, $B_1$, $\Delta_z$, and $r$ are defined in Lemma 4.1. Furthermore, if $N = \lfloor (\log_\alpha T)/2 \rfloor$ and $S_t = \lceil 2T/\log_\alpha T \rceil$ for each $1 \leq t \leq N$ where $\alpha > 1$, we have*

$$\mathbb{E}[\|\nabla f(\hat{x}_T)\|^2] \leq \frac{\alpha^2 W_1^1}{2\ln\alpha} \frac{\ln T}{T^{3/2}} + \frac{(\alpha C_0 + C_2)}{2\ln\alpha} \frac{\ln T}{T} + \frac{(\Delta_z + C_1)}{2m\ln\alpha} \frac{\ln T}{\sqrt{T}} + \frac{\alpha^2 MB_1G^2}{2\ln\alpha} \frac{\ln T}{\sqrt{T}}. \tag{8}$$

When $N = 1$, $m \leq \eta_i^t \leq M$ and the bound (7) reduces to $\mathbb{E}[\|\nabla f(\hat{x}_T)\|^2] \leq \mathcal{O}(\frac{1}{T} + \frac{1}{mT} + M)$. If, in particular, $m$ and $M$ are of order $\mathcal{O}(1/\sqrt{T})$, then we can derive the optimal convergence for constant bandwidth step-sizes, comparable to the literature for constant step-sizes (Yan et al., 2018; Liu et al., 2020; Mai and Johansson, 2020; Defazio, 2020).

It is not easy to explicitly minimize the right-hand-side of (7) with respect to $N$. However, $N = \lfloor (\log_\alpha T)/2 \rfloor$ attempts to balance the last two terms and appears to be a good choice in practice. The theorem (see (8)) establishes an $\mathcal{O}(\ln T/\sqrt{T})$ rate under step-decay bandwidth step-size. If $M = m$, which means that the step-size follows the boundary functions, we get a near-optimal (up to $\ln T$) rate for stochastic momentum with a step-decay step-size on non-convex problems. We believe that this is the first near-optimal rate for stochastic momentum with step-decay step-size. The next result shows how an exponentially increasing stage-length allow to sharpen this guarantee even further.

**Theorem 4.3.** *Suppose the same setting as Lemma 4.1. Consider Algorithm 2, if the functions* $\delta(t) = 1/\alpha^{t-1}$ *with* $\alpha > 1$, $S_t = \lceil S_0 \alpha^{t-1} \rceil$ *with* $S_0 = \sqrt{T}$, *we have*

$$\mathbb{E}[\|\nabla f(\hat{x}_T)\|^2] \leq \mathcal{O}\left( \frac{W_1^1}{T^{3/2}} + \frac{C_0}{T} + \frac{\Delta_z}{m\sqrt{T}} + \frac{C_1}{m\sqrt{T}} + \frac{MB_1 G^2}{\sqrt{T}} \right).$$

The stage length $S_t$ in Theorem 4.3 increases exponentially from $S_1 = \lceil \sqrt{T} \rceil$ over $N = \lfloor \log_\alpha((\alpha - 1)\sqrt{T} + 1) \rfloor$ stages, resulting in an $\mathcal{O}(1/\sqrt{T})$ optimal rate for SGDM under the bandwidth-based step-decay scheme. This removes the $\ln T$ term of Theorem 8. To the best of our knowledge, this work is the first that is able to achieve an optimal rate for stochastic momentum with step-decay step-size in a general non-convex setting.

**Remark 4.4.** *(**Better convergence than Liu et al. (2020)**) We notice that reference Liu et al. (2020) analyzes multi-stage momentum and obtains the bound*

$$\mathbb{E}[\|\nabla f(\tilde{x})\|^2] \leq \mathcal{O}\left( \frac{f(x_1) - f^*}{N} + \frac{\sigma L \sum_{t=1}^N \eta^t}{N} \right). \tag{9}$$

*Here, $\tilde{x}$ is a uniformly sampled iterate (unlike our results, which favour later iterates) and $N$ is the number of stages. The result uses a time-varying momentum parameter, whose value is determined by the step-size $\eta^t$, and also assumes an inverse relationship between the step-size and stage-length, i.e. that $\eta^t S_t$ is constant. Hence, $N$ is of $\mathcal{O}(\log_\alpha T)$ and the convergence guarantee in (9) is of $\mathcal{O}(1/\log_\alpha T)$, which is far worse than the rate of Theorem 4.2 and the optimal rate of Theorem 4.3.*

## 5 NUMERICAL EXPERIMENTS

In this section, we design and evaluate several specific step-size policies that belong to the bandwidth-based family. We consider SGD with and without momentum, and compare their performances on neural network training tasks on the CIFAR10 and CIFAR100 datasets.

### 5.1 BASELINES AND PARAMETER SELECTION FOR THE BANDWIDTH STEP-SIZES

The bandwidth framework allows for a unified and streamlined (worst-case) analysis of all step-size policies that lie in the corresponding band. Within this family, the band gives a lot of freedom in crating innovative step-size policies with additional advantages. In particular, we will design a number of step-size policies that add periodic perturbations to a baseline step-size, attempting to both escape bad local minima and to improve the local convergence properties.

The step-decay bandwidth step-sizes divide the total number of iterations into a small number of stages, in which the boundary functions are constant. The width of the band are determined by the constants $m$ and $M$. We will explore step-sizes that add a decreasing perturbation within each stage, starting at the upper band at the beginning of the stage, ending at the lower bound at the end of the stage, and decaying as $1/i$, $1/\sqrt{i}$, linearly or according to a cosine function. As baseline, we consider the step-decay step-size that follows the lower boundary function $m\delta(t)$. To use the same maximum value for the bandwidth step-sizes, we do not add any perturbation in the first stage; cf. Figure 5.

For the $1/\sqrt{t}$-band, on the other hand, stages correspond to epochs and perturbing the step-size within a stage would be too frequent and lead to bias. Rather, we choose to add similar perturbations

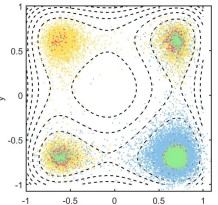
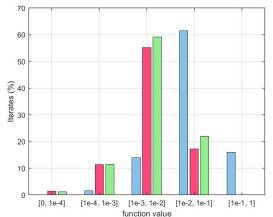

Figure 2: The scatter plots (left); function value distribution around global minima (middle)

Table 1: The percentage (%) of the final iterates (10000 runs) close to each local minima

| | constant | | step-decay | |
|---|---|---|---|---|
| | small ▉ | large ▉ | baseline ▉ | linear ▉ |
| ① | **29.61** | 0.12 | 0.40 | 0.14 |
| ② | 24.66 | 3.45 | 6.89 | 2.95 |
| ③ | 25.13 | 3.28 | 7.55 | 2.99 |
| ④ | 20.60 | **93.15** | **85.16** | **93.92** |

as for the step-decay band, but adjust the perturbation between stages. In our experiments, the two step-size policies perform roughly the same number of periods of perturbations over the training set. As baseline, we consider the step-size $\eta_i^t = m/\sqrt{t}$. In all experiments, the hyper-parameters (e.g., $m$ and $M$) have been determined using grid search, see Section E.2 for details.

## 5.2 NON-MONOTONIC SCHEDULE HELPS TO ESCAPE LOCAL MINIMA

To demonstrate the potential benefits of bandwidth-based non-monotonic step-size schedules, we consider the toy example (see Section E.3 for details and further results) from (Shi et al., 2020), which is non-convex and has four local minima[2]; see Figure 2. We then compare the final iterates of SGD with constant step-sizes (both large and small), step-decay, and a bandwidth-based step-decay step size which we call *linear-mode* (illustrated in Figure 5). As shown in Figure 2, a large constant step-size more easily escapes the bad local minima to approach the global minimum at $(0.7, -0.7)$ than a small constant step-size. However, with a large constant step-size, the final iterates are scattered and end up far from the global minimum, which also has been observed in Figure 5 of (Shi et al., 2020). Therefore, we have to reduce the step-size at some points to reduce the error. This is exactly the intuition of step-decay step-size. As shown in Figure 2, the scatter plots of SGD with *step-decay* (red) and *step-decay with linear-mode* (green) are more concentrated around the global minimum than the constant step-sizes.

To quantify the ability of different step-sizes to escape the local minima, Table 1 reports the percentage of the final iterates under the different step-size policies that are close to each minima. We can see that the ability of the step-decay policy (named *baseline*) to escape the local minima is slightly worse than the large constant step-size, but Figure 2 shows that the variance of the near-optimal iterates is reduced significantly. In a similar way, we can see that *linear-mode* not only improves the ability to escape the local minima, but also produces final iterates that are more concentrated around the global optimum. Hence, it appears (at least in this example) that non-monotonic step-size schedules allow SGD to escape local minima and produce final iterates of high quality.

## 5.3 NUMERICAL RESULTS ON CIFAR10 AND CIFAR100

To illustrate the practical performance of the bandwidth-based step-sizes, we choose the well-known CIFAR10 and CIFAR100 (Krizhevsky, 2009) image classification datasets. We consider the benchmark experiments of CIFAR10 on ResNet-18 (He et al., 2016) and CIFAR100 on a $28 \times 10$ wide residual network (WRN-28-10) (Zagoruyko and Komodakis, 2016), respectively. All the experiments are repeated 5 times to eliminate the influence of randomness.

We begin by evaluating our step-sizes for SGD. The left column of Figure 3 present the results of the $1/\sqrt{t}$-band step-sizes on the two datasets. As shown in Figure 5, these stepsizes are all non-monotonic. The sudden increase in the step-size leads to a corresponding cliff-like reduction in accuracy followed by a recovery phase that consistently ends up at a better performance than in the previous stage. The three $1/\sqrt{t}$-band step-sizes achieve significant improvements compared to their baseline ($\eta_i^t = m/\sqrt{t}$), in terms of both test loss and test accuracy. Moreover, the linear-mode performs the best compared to other polynomial decaying modes. Then, the results of SGD with step-decay band (described in Section 5.1 or see Figure 5 in Appendix) on CIFAR10 and CIFAR100

---

[2]Notation: ①−③ denote the local minima at top left, top right and bottom left, respectively; and ④ denotes the global minimum at bottom right.

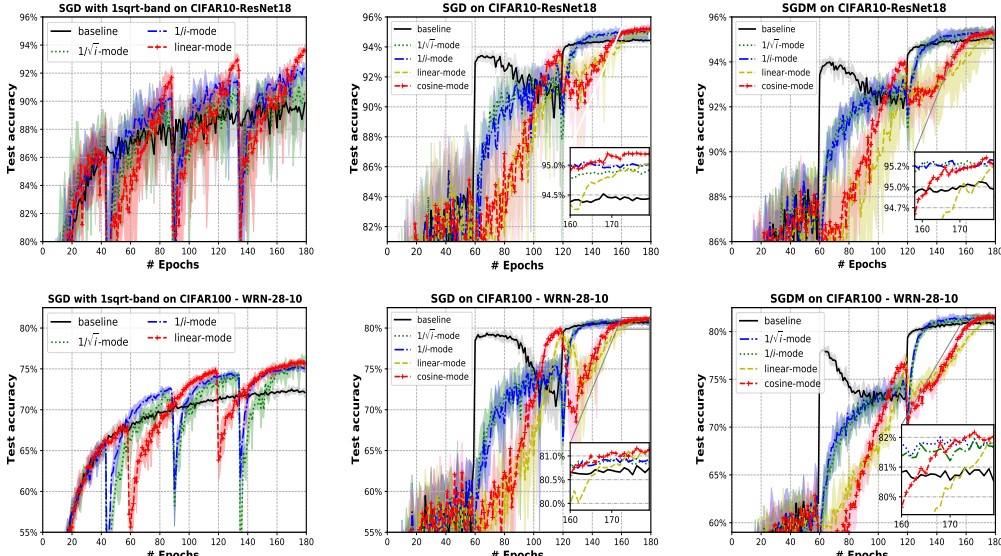

Figure 3: The test accuracy of $1/\sqrt{t}$-band (left column), Step-decay-band for SGD (middle column), step-decay band for SGDM (right column)

are given in Figure 3 (middle column), respectively. At the final stage, the bandwidth step-sizes improve both test loss (see Figures 6 and 7 in Appendix) and test accuracy compared to the baseline. In particular, the cosine-mode performs the best on this problem. In the second stage, baseline methods have a sharp boost. Our guess is that the noise accumulates quickly under a relatively large constant step-size. But this phenomenon is only temporary. When we drop the step-size in the third stage, the performance improves.

Next, we evaluate the performance of step-decay bandwidth step-sizes on SGDM. The results are reported in Figure 3 (right column). The first observation from Figure 3 (right column) is that the step-decay bandwidth step-sizes also work well for SGDM, and that again, the cosine-mode performs better than the others. Another interesting observation is that the performance of vanilla SGD with cosine-mode (red) in Figure 3 is comparable to (even better than) SGDM with the baseline step-decay step-size (black) in Figure 3. A similar conclusion can also be made on CIFAR100.

## 6 CONCLUSION

We have studied a general family of bandwidth step-sizes for non-convex optimization. The family specifies a globally decaying band in which the actual step-size is allowed to vary, and includes both stage-wise and continuously decaying step-size policies as special cases. We have derived convergence rate guarantees for SGD and SGDM under all step-size policies in two important classes of bandwidth step-sizes ($1/\sqrt{t}$ and step-decay), some of which are optimal. Our results provide theoretical guarantees for several popular "cyclical" step-sizes (Loshchilov and Hutter, 2017; Smith, 2017), as long as they are tuned to lie within our bands. We have also designed a number of novel step-sizes that add periodic perturbations to the global trend in order to escape bad local minima and to improve the local convergence properties. These step-sizes were shown to have superior practical performance in neural network training tasks on the CIFAR data set.

In the analysis of SGDM, we assume that the stochastic gradient is bounded (see Assumption 2(b)). It is interesting to see how to relax this assumption in some special cases, for example, when the step-size is constant throughout each stage. It would also be interesting to see if the bandwidth framework could be specialized to a more narrow class of step-sizes, for which we can provide even stronger convergence rates.

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

## A  CONVERGENCE FOR BANDWIDTH POLYNOMIAL-DECAY STEP-SIZES

Our first more specific result considers the $1/\sqrt{t}$ bandwidth step-size with fixed stage length.

**Theorem A.1.** *Under Assumptions 1 and 2(a), and assume that there exists a constant $\Delta_0 > 0$ such that $\mathbb{E}[f(x_1^t) - f^*] \leq \Delta_0$ for each $t \geq 1$. If the step-size $\eta_i^t \leq 1/((\rho + 1)L)$, the stage length $S_t = S \geq 1$, and the boundary function $\delta(t) = 1/\sqrt{t}$ for each $1 \leq t \leq N$, we have*

$$\mathbb{E}[\|\nabla f(\hat{x}_T)\|^2] \leq \frac{3\Delta_0}{m} \cdot \frac{1}{\sqrt{ST}} + \frac{3M^2 L\sigma}{2m} \cdot \sqrt{\frac{S}{T}}. \tag{10}$$

Theorem A.1 shows how multi-stage SGD with polynomial-decay bandwidth step-sizes converges to a stationary point. In the extreme case that $S = 1$, the step-size reduces to $m/\sqrt{t} \leq \eta_1^t \leq M/\sqrt{t}$ and our result is comparable to the non-asymptotic optimal rate derived for $m = M = \eta_0$ in (Wang et al., 2021, Theorem 3.5).

**Multi-stage vs traditional** $1/\sqrt{t}$ **step-size** ($S = 1$) In general, during the initial iterations when the first term of (10) dominates the error bound, the multi-stage technique can accelerate the convergence by a larger step-size and longer inner-loop $S$. However, a large $S$ will make the error bound worse when the noise term begins to dominate the bound. The next theorem analyzes an algorithm with a decreasing stage length.

**Theorem A.2.** *Under Assumptions 1 and 2(a), and assume that there exists a constant $\Delta_0 > 0$ such that $\mathbb{E}[f(x_1^t) - f^*] \leq \Delta_0$ for each $t \geq 1$. If the step-size $\eta_i^t \leq 1/((\rho + 1)L)$, the stage length $S_t = \lceil S_0/\sqrt{t} \rceil$ with $S_0 = \sqrt{T}$, and $\delta(t) = 1/\sqrt{t}$ for each $1 \leq t \leq N$, we have*

$$\mathbb{E}[\|\nabla f(\hat{x}_T)\|^2] \leq \left(2\Delta_0 + \frac{M^2 L\sigma}{3 - 2\sqrt{2}}\right) \cdot \frac{1}{m\sqrt{T}}.$$

**Schedule of Theorem A.2 vs Chen et al. (2019)** The theorem establishes an optimal rate for multi-stage SGD with $1/\sqrt{t}$ bandwidth step-size. Note that Chen et al. (2019) also analyzes a stagewise algorithm with varying stage length, but their step-size decays as $1/t$ and stage length increases with $t$. An important novelty with our result is that it uses a long initial stage, $S_1 = \sqrt{T}$ while a large stage length in Chen et al. (2019) requires a small initial step-size (of $\mathcal{O}(1/\sqrt{T})$). Figure 4 illustrates the performance of different step-size policies: 1) $1/\sqrt{t}$ with $S_t = 1$; 2) $1/\sqrt{t}$ with $S_t = \lceil n/b \rceil$ where $n$ is total sample size and $b$ is the batch size; 3) $1/\sqrt{t}$ with time-decreasing $S_t = \lceil S_0/\sqrt{t} \rceil$ and $S_0 = \sqrt{T}$; 4) and $1/t$ step-size with $S_t = S_0 t$ and $S_0 = 12$ from Chen et al. (2019). We can see that the step-size policies proposed in Theorem A.2 are more stable and perform the best.

For completeness, we also compare the performance of step-size schedules proposed by Theorems 3.1 and 3.3 in Figure 4 (right). Although step-decay with time-increasing stage length has a superior theoretical convergence guarantee, constant stage length performs better in this particular example.

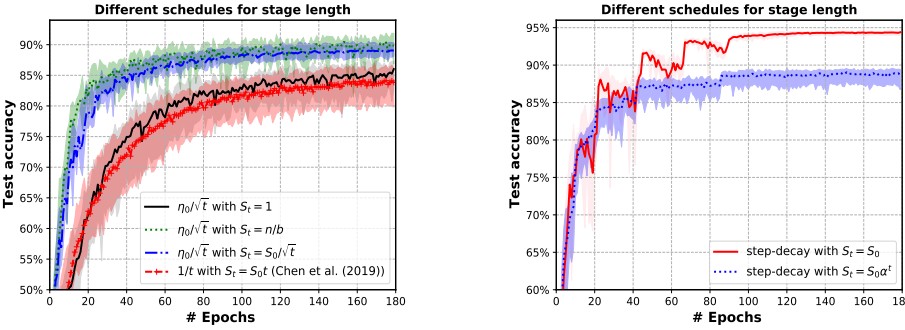

Figure 4: SGD - CIFAR10 - ResNet18

# B    PROOFS OF LEMMA AND THEOREMS IN SECTION 3

**Lemma B.1.** *Suppose that Assumption 1 and Assumption 2(a) hold. If we run the Algorithm 1 with $T > 1$ and $\eta_i^t \leq 1/((\rho + 1)L)$, we have*

$$\mathbb{E}[\|\nabla f(\hat{x}_T)\|^2] \leq \frac{1}{\displaystyle\sum_{t=1}^{N} S_t \delta^{-1}(t)} \left( \sum_{t=1}^{N} \frac{2(\mathbb{E}[f(x_1^t)] - \mathbb{E}[f(x_1^{t+1})])}{m\delta(t)^2} + \frac{M^2 L\sigma}{m} T \right). \quad (11)$$

*Proof.* **(of Lemma B.1)** The $L$-smoothness of $f$ (see Assumption 1), i.e., $\|\nabla f(x) - \nabla f(y)\| \leq L \|x - y\|$ for all $x, y \in \text{dom}(f)$ implies that

$$f(x) + \langle \nabla f(x), y - x \rangle - \frac{L}{2} \|x - y\|^2 \leq f(y) \leq f(x) + \langle \nabla f(x), y - x \rangle + \frac{L}{2} \|x - y\|^2. \quad (12)$$

Applying the $L$-smoothness property of $f$ and recalling Algorithm 1 at current iterate $x_i^t$, we have

$$f(x_{i+1}^t) \leq f(x_i^t) + \left\langle \nabla f(x_i^t), x_{i+1}^t - x_i^t \right\rangle + \frac{L}{2} \left\| x_{i+1}^t - x_i^t \right\|^2$$

$$\leq f(x_i^t) - \eta_i^t \left\langle \nabla f(x_i^t), g_i^t \right\rangle + \frac{(\eta_i^t)^2 L}{2} \left\| g_i^t \right\|^2.$$

Taking conditional expectation of $\mathcal{F}_i^t$ on the above inequality and due to the unbiased estimator $g_i^t$ such that $\mathbb{E}[g_i^t \mid \mathcal{F}_i^t] = \nabla f(x_i^t)$, we obtain that

$$\mathbb{E}[f(x_{i+1}^t)|\mathcal{F}_i^t] \leq f(x_i^t) - \eta_i^t \left\| \nabla f(x_i^t) \right\|^2 + \frac{(\eta_i^t)^2 L}{2} \mathbb{E}[\|g_i^t\|^2 |\mathcal{F}_i^t]. \quad (13)$$

By Assumption 2(a) that $\mathbb{E}[\|g_i^t - \nabla f(x_i^t)\|^2 \mid \mathcal{F}_i^t] \leq \rho \|\nabla f(x_i^t)\|^2 + \sigma$, we have

$$\mathbb{E}[\|g_i^t\|^2 \mid \mathcal{F}_i^t] = \mathbb{E}[\|g_i^t - \nabla f(x_i^t) + \nabla f(x_i^t)\|^2]$$

$$= \mathbb{E}[\|g_i^t - \nabla f(x_i^t)\|^2] + \mathbb{E}[\|\nabla f(x_i^t)\|^2] \leq (\rho + 1) \|\nabla f(x_i^t)\|^2 + \sigma. \quad (14)$$

Then incorporating the above inequality into (13) gives

$$\mathbb{E}[f(x_{i+1}^t)|\mathcal{F}_i^t] \leq f(x_i^t) + \left( -\eta_i^t + \frac{(\eta_i^t)^2 L(\rho + 1)}{2} \right) \left\| \nabla f(x_i^t) \right\|^2 + \frac{(\eta_i^t)^2 L\sigma}{2}. \quad (15)$$

If step-size $\eta_i^t \leq 1/((\rho + 1)L)$, we have $-\eta_i^t + \frac{(\eta_i^t)^2 L(\rho+1)}{2} \leq -\eta_i^t/2$. For any $t \geq 1$, the inequality (15) can be estimated as:

$$\frac{\eta_i^t}{2} \left\| \nabla f(x_i^t) \right\|^2 \leq f(x_i^t) - \mathbb{E}[f(x_{i+1}^t)|\mathcal{F}_i^t] + \frac{(\eta_i^t)^2 L\sigma}{2}. \quad (16)$$

Applying the assumption of step-size that $\eta_i^t = n(t,i)\delta(t)$ with $m \leq n(t,i) \leq M$ for all $t \in \{1, 2, \cdots, N\}$ to (16) gives

$$\frac{m\delta(t)}{2} \left\| \nabla f(x_i^t) \right\|^2 \leq f(x_i^t) - \mathbb{E}[f(x_{i+1}^t)|\mathcal{F}_i^t] + \frac{M^2 L\sigma}{2} \delta(t)^2, \quad (17)$$

and dividing $m\delta(t)^2/2$ into the both sides, we have

$$\delta^{-1}(t) \left\| \nabla f(x_i^t) \right\|^2 \leq \frac{2 \left( f(x_i^t) - \mathbb{E}[f(x_{i+1}^t)|\mathcal{F}_i^t] \right)}{m\delta(t)^2} + \frac{M^2 L\sigma}{m}. \quad (18)$$

Recalling the output $\hat{x}_T$ of Algorithm 1 and then taking expectation, we have

$$\mathbb{E}[\|\nabla f(\hat{x}_T)\|^2] = \frac{1}{\sum_{t=1}^{N} S_t \delta^{-1}(t)} \sum_{t=1}^{N} \delta^{-1}(t) \cdot \sum_{i=1}^{S_t} \mathbb{E} \left[ \left\| \nabla f(x_i^t) \right\|^2 \right]. \quad (19)$$

Applying (18) recursively from $i = 1$ to $S_t$ and using the fact that $x_1^{t+1} = x_{S_t+1}^t$, the sum of (19) for $t \geq 1$ can be estimated as

$$\sum_{t=1}^{N} \delta^{-1}(t) \sum_{i=1}^{S_t} \mathbb{E}\left[\|\nabla f(x_i^t)\|^2\right] \leq \sum_{t=1}^{N} \frac{2(\mathbb{E}[f(x_1^t)] - \mathbb{E}[f(x_1^{t+1})])}{m\delta(t)^2} + \frac{M^2 L \sigma}{m} T. \qquad (20)$$

Then plugging (20) into (19), we get

$$\mathbb{E}[\|\nabla f(\hat{x}_T)\|^2] \leq \frac{1}{\sum_{t=1}^{N} S_t \delta^{-1}(t)} \left(\sum_{t=1}^{N} \frac{2(\mathbb{E}[f(x_1^t)] - \mathbb{E}[f(x_1^{t+1})])}{m\delta(t)^2} + \frac{M^2 L \sigma}{m} T\right)$$

as desired.

$\square$

*Proof.* (**of Theorem A.1**) In this case, the step size $\eta_i^t$ satisfies that $m/\sqrt{t} \leq \eta_i^t \leq M/\sqrt{t}$, where $\delta(t) = 1/\sqrt{t}$. By the definition of $\delta(t)$, we have

$$\sum_{t=1}^{N} \frac{1}{\delta(t)} \geq \int_{t=0}^{N} \frac{1}{\delta(t)} dt = \frac{2N^{\frac{3}{2}}}{3}. \qquad (21)$$

Applying the assumption that $\mathbb{E}[f(x_1^t) - f^*] \leq \Delta_0$ where $f^* = \min_x f(x)$ and the definition of $\delta(t)$, we have

$$\sum_{t=1}^{N} \frac{2(\mathbb{E}[f(x_1^t)] - \mathbb{E}[f(x_{S+1}^t)])}{m\delta(t)^2} \leq \frac{2}{m} \sum_{t=1}^{N} t(\mathbb{E}[f(x_1^t) - f^*] - \mathbb{E}[f(x_1^{t+1}) - f^*])$$

$$\leq \frac{2}{m}\left(\Delta_0 + \sum_{t=2}^{N} \mathbb{E}[f(x_1^t) - f^*]\right) \leq \frac{2N\Delta_0}{m}. \qquad (22)$$

Under Assumptions 1 and 2(a) and $\eta_i^t < 1/((\rho + 1)L)$, thus Lemma B.1 holds. Incorporating these inequalities into Lemma B.1 gives

$$\mathbb{E}[\|\nabla f(\hat{x}_T)\|^2] \leq \frac{1}{S \sum_{t=1}^{N} \delta^{-1}(t)} \left[\sum_{t=1}^{N} \frac{2(\mathbb{E}[f(x_1^t)] - \mathbb{E}[f(x_1^{t+1})])}{m\delta(t)^2} + \frac{SM^2 L \sigma}{m} T\right]$$

$$\leq \frac{3}{2SN^{3/2}} \left[\frac{2N\Delta_0}{m} + \frac{M^2 L \sigma}{m} T\right]$$

$$\leq \frac{3\Delta_0}{m} \cdot \frac{1}{\sqrt{ST}} + \frac{3M^2 L \sigma}{2m} \cdot \sqrt{\frac{S}{T}}.$$

Then the proof is finished.

$\square$

*Proof.* (**of Theorem A.2**) In this theorem, we consider SGD with the $1/\sqrt{t}$ bandwidth step-size, i.e., $\frac{m}{\sqrt{t}} \leq \eta_i^t \leq \frac{M}{\sqrt{t}}$ for $1 \leq t \leq N$, where the stage length $S_t = \lceil S_0/\sqrt{t} \rceil$ with $S_0 = \sqrt{T}$, and the boundary function $\delta(t) = 1/\sqrt{t}$. By the relationship that $\sum_{t=1}^{N} S_t = T$ and $x \leq \lceil x \rceil \leq x + 1$ for all $x$, we get that

$$(3 - 2\sqrt{2})T \leq N \leq \left(\frac{\sqrt{T}}{2} + 1\right)^2. \qquad (23)$$

Under Assumptions 1 and 2(a) and $\eta_i^t < 1/((\rho + 1)L)$, thus Lemma B.1 holds. Following the same process as Theorem A.1, the inequality (22) also holds, that is

$$\sum_{t=1}^{N} \frac{2(\mathbb{E}[f(x_1^t)] - \mathbb{E}[f(x_{S+1}^t)])}{m\delta(t)^2} \leq \frac{2N\Delta_0}{m}. \qquad (24)$$

Then incorporating the above inequalities to Lemma B.1, we have

$$
\begin{aligned}
\mathbb{E}[\|\nabla f(\hat{x}_T)\|^2] &\le \frac{1}{\sum_{t=1}^N S_t \delta^{-1}(t)} \left[ \frac{2N\Delta_0}{m} + \frac{M^2 L\sigma}{m} T \right] \\
&\le \frac{1}{\sum_{t=1}^N \lceil S_0/\sqrt{t} \rceil \cdot \sqrt{t}} \left[ \frac{2N\Delta_0}{m} + \frac{M^2 L\sigma}{m} \cdot T \right] \\
&\le \frac{1}{S_0 N} \left[ \frac{2N\Delta_0}{m} + \frac{M^2 L\sigma}{m} \cdot T \right] \\
&\le \frac{2\Delta_0}{m S_0} + \frac{M^2 L\sigma}{(3-2\sqrt{2})m S_0} = \frac{1}{m} \left( 2\Delta_0 + \frac{M^2 L\sigma}{3-2\sqrt{2}} \right) \cdot \frac{1}{\sqrt{T}},
\end{aligned}
$$

which concludes the proof. $\qquad\square$

*Proof.* (**of Theorem 3.1**) In this case, the step-size $\eta_i^t$ exponentially decays every $S$ iterations. By the definition of $\eta_i^t$, we have $\delta(t) = 1/\alpha^{t-1}$ for all $1 \le t \le N$. The sum of $1/\delta(t)$ can be estimated as

$$
\sum_{t=1}^N \frac{1}{\delta(t)} = \sum_{t=1}^N \alpha^{t-1} = \frac{\alpha^N - 1}{\alpha - 1}.
$$

Recalling the assumption that $\mathbb{E}[f(x_1^t) - f^*] \le \Delta_0$ for all $t \ge 1$, we have

$$
\begin{aligned}
\sum_{t=1}^N \frac{2(\mathbb{E}[f(x_1^t)] - \mathbb{E}[f(x_1^{t+1})])}{m\delta(t)^2} &\le \frac{2}{m} \sum_{t=1}^N \alpha^{2(t-1)} (\mathbb{E}[f(x_1^t) - f^*] - \mathbb{E}[f(x_1^{t+1}) - f^*]) \\
&\le \frac{2}{m} \left( (\alpha^2 - 1) \sum_{t=2}^N \alpha^{2(t-2)} \mathbb{E}[f(x_1^t) - f^*] + \Delta_0 \right) \\
&\le \frac{2\alpha^{2(N-1)} \Delta_0}{m}.
\end{aligned}
$$

Under Assumptions 1 and 2(a) and $\eta_i^t < 1/((\rho+1)L)$, thus Lemma B.1 holds. Then applying the result of Lemma B.1 and incorporating the above inequalities into Lemma B.1 gives

$$
\begin{aligned}
\mathbb{E}[\|\nabla f(\hat{x}_T)\|^2] &\le \frac{1}{S \sum_{t=1}^N \delta_1^{-1}(t)} \left( \sum_{t=1}^N \frac{2(\mathbb{E}[f(x_1^t)] - \mathbb{E}[f(x_{S+1}^t)])}{m\delta(t)^2} + \frac{SM^2 L\sigma}{m} \cdot \sum_{t=1}^N \frac{\delta_2(t)^2}{\delta(t)^2} \right) \\
&\le \frac{\alpha-1}{S(\alpha^N-1)} \left( \frac{2\alpha^{2(N-1)}\Delta_0}{m} + \frac{M^2 L\sigma}{m} SN \right).
\end{aligned}
$$

Substituting the specific values of $N = \lfloor (\log_\alpha T)/2 \rfloor$, $S = \lceil 2T/\log_\alpha T \rceil$ into the above inequality, we have

$$
\begin{aligned}
\mathbb{E}[\|\nabla f(\hat{x}_T)\|^2] &\le \frac{(\alpha-1)}{S(\alpha^N-1)} \left( \frac{\Delta_0 \alpha^{2(N-1)}}{m} + \frac{M^2 L\sigma}{m} SN \right) \\
&\le \left( \frac{\Delta_0}{2\alpha m} + \frac{\alpha M^2 L\sigma}{2m} \right) \frac{(\alpha-1)\log_\alpha T}{\sqrt{T} - \alpha}.
\end{aligned}
$$

then transform the base $\log_\alpha$ to the natural logarithm $\ln$, we can get the desired result. $\qquad\square$

*Proof.* (**of Theorem 3.3**) If $S_0 = \sqrt{T}$ and $S_t = \lceil S_0 \alpha^{t-1} \rceil$, by $\sum_{t=1}^N S_t = T$, the stage length $N$ is $\lfloor \log_\alpha((\alpha-1)\sqrt{T}+1) \rfloor$. In this case, we have $\delta(t) = 1/\alpha^{t-1}$ for each $t \in [N]$. Under Assumptions 1 and 2(a) and $\eta_i^t < 1/((\rho+1)L)$, thus Lemma B.1 holds. Before applying the result of Lemma B.1, we first give the following estimations:

$$
\sum_{t=1}^N S_t \delta^{-1}(t) \ge \sum_{t=1}^N S_0 \alpha^{t-1} \cdot \alpha^{t-1} = \sqrt{T} \cdot \frac{1 - \alpha^{2N}}{1 - \alpha^2} = \frac{(\alpha-1)T^{\frac{3}{2}} + 2T}{\alpha + 1} \tag{25}
$$

and also

$$\sum_{t=1}^{N} \frac{2(\mathbb{E}[f(x_1^t)] - \mathbb{E}[f(x_1^{t+1})])}{m\delta(t)^2} \leq \frac{2}{m}\left((\alpha^2 - 1)\sum_{t=2}^{N} \alpha^{2(t-2)}\mathbb{E}[f(x_1^t) - f^*] + \Delta_0\right)$$

$$\leq \frac{2\alpha^{2(N-1)}\Delta_0}{m} = \frac{2((\alpha-1)\sqrt{T}+1)^2\Delta_0}{\alpha^2 m}.$$

Applying these results into Lemma B.1, we have

$$\mathbb{E}[\|\nabla f(\hat{x}_T)\|^2] \leq \frac{\alpha+1}{(\alpha-1)T^{\frac{3}{2}} + 2T}\left(\frac{2((\alpha-1)\sqrt{T}+1)^2\Delta_0}{\alpha^2 m} + \frac{M^2 L\sigma}{m}\cdot T\right)$$

$$\leq \frac{\alpha+1}{\alpha-1}\left(\frac{2\Delta_0}{m} + \frac{M^2 L\sigma}{m}\right)\frac{1}{\sqrt{T}} + \mathcal{O}\left(\frac{1}{T}\right).$$

Thus the proof is complete. $\qquad\qquad\qquad\qquad\qquad\qquad\qquad\qquad\qquad\qquad\qquad\qquad\qquad\quad$ □

In Lemma B.1, we provide a unified analysis framework for bandwidth step-sizes which are independent on the current random information. Recently, there are some interesting non-monotonic step-sizes, e.g., the trust-region-ish algorithm (Curtis et al., 2019) and stochastic Polyak step-sizes (Loizou et al., 2021), which can also be regarded to be in a band, but those are related to the current stochastic information. We provide a unified framework for these kind of step-sizes below.

**Lemma B.2.** *Under the same conditions as in Lemma B.1, we assume that the step-size $\eta_i^t$ satisfies $m\delta(t) \leq \eta_i^t \leq M\delta(t)$ and is dependent on the current random information. If $m \leq \frac{4}{L(\rho+1)}$ and $m \leq M \leq \tilde{M} := \frac{-\rho+\sqrt{\rho^2+2(\rho+1)(2+\rho)mL}}{L(\rho+1)}$, we have*

$$\mathbb{E}[\|\nabla f(\tilde{x}_T)\|^2] \leq \frac{1}{\psi_0 \Sigma_T}\left(\frac{\Delta_0}{\delta(N)^2} + (M-m)\sigma\Sigma_T + \frac{LM^2\sigma}{2}T\right) \tag{26}$$

*where $\Sigma_T = \sum_{t=1}^{N}\sum_{i=1}^{S_t}\delta(t)^{-1}$ and $\psi_0 = (2+\rho)m - \rho M - \frac{LM^2}{2}(\rho+1)$.*

*Proof.* We consider the step-size $\eta_i^t$ is depended on the current random information and $m\delta(t) \leq \eta_i^t \leq M\delta(t)$. By the $L$-smoothness of $f$, that is $\|\nabla f(x) - \nabla f(y)\| \leq L\|x-y\|$, implies that

$$f(y) \leq f(x) + \langle \nabla f(x), y - x \rangle + \frac{L}{2}\|x-y\|^2 \tag{27}$$

Let $x = x_i^t$ and $y = x_{i+1}^t$, then

$$f(x_{i+1}^t) \leq f(x_i^t) - \langle \nabla f(x_i^t), \eta_i^t g_i^t \rangle + \frac{L}{2}\|\eta_i^t g_i^t\|^2 \tag{28}$$

Next we turn to estimate the product term $-\langle \nabla f(x_i^t), \eta_i^t g_i^t \rangle$.

$$-\langle \nabla f(x_i^t), \eta_i^t g_i^t \rangle = \eta_i^t\left(\|g_i^t - \nabla f(x_i^t)\|^2 - \|g_i^t\|^2 - \|\nabla f(x_i^t)\|^2\right)$$

$$\leq M\delta(t)\|g_i^t - \nabla f(x_i^t)\|^2 - m\delta(t)\|g_i^t\|^2 - m\delta(t)\|\nabla f(x_i^t)\|^2. \tag{29}$$

Then taking conditional expectation on $\mathcal{F}_i^t$ to (28) and applying Assumption 2(a) and (14), we have

$$
\begin{aligned}
\mathbb{E}[f(x_{i+1}^t) \mid \mathcal{F}_i^t] &\leq f(x_i^t) - m\delta(t) \left\| \nabla f(x_i^t) \right\|^2 - m\delta(t)\mathbb{E}[\|g_i^t\|^2 \mid \mathcal{F}_i^t] + M\delta(t)\mathbb{E}[\|g_i^t - \nabla f(x_i^t)\|^2 \mid \mathcal{F}_i^t] \\
&\quad + \frac{LM^2\delta(t)^2}{2} \mathbb{E}[\|g_i^t\|^2 \mid \mathcal{F}_i^t] \\
&\leq f(x_i^t) - m\delta(t) \left\| \nabla f(x_i^t) \right\|^2 - m\delta(t) \left( \mathbb{E}[\|g_i^t - \nabla f(x_i^t)\|^2] + \left\| \nabla f(x_i^t) \right\|^2 \right) \\
&\quad + M\delta(t)\mathbb{E}[\|g_i^t - \nabla f(x_i^t)\|^2 \mid \mathcal{F}_i^t] + \frac{LM^2\delta(t)^2}{2} \left( \mathbb{E}[\|g_i^t - \nabla f(x_i^t)\|^2] + \left\| \nabla f(x_i^t) \right\|^2 \right) \\
&\leq f(x_i^t) - \left( 2m\delta(t) - \frac{LM^2\delta(t)^2}{2} \right) \left\| \nabla f(x_i^t) \right\|^2 + \left( M\delta(t) - m\delta(t) + \frac{LM^2\delta(t)^2}{2} \right) \mathbb{E}[\|g_i^t - \nabla f(x_i^t)\|^2] \\
&\leq f(x_i^t) - \left( 2m - \frac{LM^2\delta(t)}{2} - \rho \left( M - m + \frac{LM^2\delta(t)}{2} \right) \right) \delta(t) \left\| \nabla f(x_i^t) \right\|^2 \\
&\quad + \left( M - m + \frac{LM^2\delta(t)}{2} \right) \delta(t)\sigma
\end{aligned}
$$

where by $M \geq m$ we have $M\delta(t) - m\delta(t) + \frac{LM^2\delta(t)^2}{2} > 0$. Let $\psi := 2m - \frac{LM^2\delta(t)}{2} - \rho \left( M - m + \frac{LM^2\delta(t)}{2} \right)$. We know that $\delta(t) \leq 1$, then

$$
\psi \geq (2 + \rho)m - \rho M - \frac{LM^2}{2}(\rho + 1). \tag{30}
$$

Let $\psi_0 = (2 + \rho)m - \rho M - \frac{LM^2}{2}(\rho + 1)$. By solving the quadratic expression of $M$ in $\psi_0$, if

$$
m \leq M < \tilde{M} := \frac{-\rho + \sqrt{\rho^2 + 2(\rho+1)(2+\rho)mL}}{L(\rho+1)} \tag{31}
$$

then we have $\psi \geq \psi_0 > 0$. To guarantee that $m \leq M$, we require that $\psi_0(m) > 0$, then $m \leq \frac{4}{L(\rho+1)}$. Recalling the output $\tilde{x}$ which is selected from $\{x_i^t\}$ for all $i \in [S_t]$ and $t \in [N]$ with probability $P_t \propto 1/\delta(t)$, we have

$$
\begin{aligned}
\mathbb{E}[\|\nabla f(\tilde{x})\|^2] &= \frac{1}{\sum_{t=1}^N S_t \delta(t)^{-1}} \sum_{t=1}^N \delta(t)^{-1} \sum_{i=1}^{S_t} \mathbb{E}[\|\nabla f(x_i^t)\|^2] \\
&\leq \frac{1}{\psi_0 \Sigma_T} \left( \frac{\sum_{t=1}^N \left( \mathbb{E}[f(x_1^t)] - \mathbb{E}[f(x_{S_t+1}^t)] \right)}{\delta(t)^2} + (M-m)\sigma \sum_{t=1}^N \sum_{i=1}^{S_t} \delta(t)^{-1} + \frac{LM^2\sigma}{2}T \right) \\
&\overset{(a)}{\leq} \frac{1}{\psi_0 \Sigma_T} \left( \frac{\Delta_0}{\delta(N)^2} + (M-m)\sigma\Sigma_T + \frac{LM^2\sigma}{2}T \right) \tag{32}
\end{aligned}
$$

where $\Sigma_T = \sum_{t=1}^N S_t \delta(t)^{-1}$ and (a) follows from the assumption that $\mathbb{E}[f(x_1^t) - f^*] \leq \Delta_0$ where $f^* = \min_x f(x)$ and $x_1^{t+1} = x_{S_t+1}^t$. $\qquad\square$

The above lemma immediately results in the following convergence results for bandwidth step-sizes which depend on the current random information.

- If $N = 1$ and $S_t = T$, that is $m \leq \eta_i^t \leq M$ and $\eta_i^t$ is related to the current random information, we have

$$
\mathbb{E}[\|\nabla f(\tilde{x})\|^2] \leq \frac{\Delta_0}{\psi_0 T} + \frac{(M-m) + LM^2/2}{\psi_0}\sigma \tag{33}
$$

In this case, the boundary function $\delta(t) = 1$, so the assumption on function value can be replaced by $f(x_0^1) - f^*$ is bounded. We can achieve an $\mathcal{O}(1/T)$ convergence rate to reach a neighborhood of the stationary point. This error bound is comparable to the results of (Loizou et al., 2021, Theorem 3.8) and (Curtis et al., 2019, Theorem 3.5).

---

**Algorithm 2** SGDM with Bandwidth-based Step-Size

---

1: **Input:** initial point $x_1^1 \in \mathbb{R}^d$, $v_1^1 = \mathbf{0}$, # iterations $T$, # stages $N$, stage length $\{S_t\}_{t=1}^N$ such that $\sum_{t=1}^N S_t = T$, momentum parameter $\beta \in (0,1)$, the sequences $\{\delta(t)\}_{t=1}^N$ and $\left\{ \{n(t,i)\}_{i=1}^{S_t} \right\}_{t=1}^N \in [m, M]$ with $0 < m \le M$

2: **for** $t = 1 : N$ **do**

3:     **for** $i = 1 : S_t$ **do**

4:         Query a stochastic gradient oracle $\mathcal{O}$ at $x_i^t$ to get a vector $g_i^t$ such that $\mathbb{E}[g_i^t \mid \mathcal{F}_i^t] = \nabla f(x_i^t)$

5:         Update step-size $\eta_i^t = n(t,i)\delta(t)$, which belongs to the interval $[m\delta(t), M\delta(t)]$

6:         $v_{i+1}^t = \beta v_i^t + (1-\beta)g_i^t$

7:         $x_{i+1}^t = x_i^t - \eta_i^t v_{i+1}^t$

8:     **end for**

9:     $v_1^{t+1} = v_{S_t+1}^t$ and $x_1^{t+1} = x_{S_t+1}^t$

10: **end for**

11: **Return:** $\hat{x}_T$ is uniformly chosen from $\left\{ x_1^{t^*}, x_2^{t^*}, \cdots, x_{S_{t^*}}^{t^*} \right\}$, where the integer $t^*$ is randomly chosen from $\{1, 2, \cdots, N\}$ with probability $P_t = \delta^{-1}(t)/(\sum_{l=1}^N \delta^{-1}(l))$

---

- If $N > 1$, $S_t = S$ and $\delta(t) = 1/\alpha^{t-1}$, we have

$$\mathbb{E}[\|\nabla f(\tilde{x})\|^2] \le \frac{\alpha-1}{\psi_0 S(\alpha^N - 1)} \left( \Delta_0 \alpha^{2(N-1)} + \frac{LM^2\sigma}{2}T \right) + \frac{(M-m)\sigma}{\psi_0}$$

$$= \frac{1}{\psi_0} \left( \frac{N\alpha^{N-1}}{T} + \frac{LM^2\sigma}{2}\frac{(\alpha-1)N}{(\alpha^N - 1)} \right) + \frac{(M-m)\sigma}{\psi_0} \tag{34}$$

Compared to the case that $N = 1$, we observe that increasing $N > 1$ can improve the error term $\frac{LM^2/2}{\psi_0}\sigma$ of (33). If $M - m \le \frac{LM^2}{2}$, i.e., $m \ge M - \frac{LM^2}{2}$, then $\frac{LM^2/2}{\psi_0}\sigma$ turns out to dominate the error bound of (33). If we increase $N$ appropriately, then the error term of (33) can be improved.

## C    PROOFS OF LEMMA AND THEOREMS IN SECTION 4

We recall the momentum scheme of Algorithm 2 below

$$v_{i+1}^t = \beta v_i^t + (1-\beta)g_i^t \tag{35}$$

$$x_{i+1}^t = x_i^t - \eta_i^t v_{i+1}^t \tag{36}$$

where $\beta \in (0,1)$. Before giving the proofs, we introduce an extra variable $z_i^t = \frac{x_i^t}{1-\beta} - \frac{\beta}{1-\beta}x_{i-1}^t$, then

$$z_i^t - x_i^t = \frac{\beta}{1-\beta}(x_i^t - x_{i-1}^t), \tag{37}$$

$$z_{i+1}^t - x_i^t = \frac{1}{1-\beta}(x_{i+1}^t - x_i^t). \tag{38}$$

However, the bandwidth-based step-size $\eta_i^t$ in our analysis is time dependent and also possibly non-monotonic, so the commonly used equalities $x_{i+1}^t = x_i^t - \eta g_i^t + \beta(x_i^t - x_{i-1}^t)$ (Yan et al., 2018) or $z_{i+1}^t = z_i^t - \eta^t g_i^t$ (see lemma 3 of Liu et al. (2020)) do not hold in our analysis. This significantly increases the level of difficulty of the analysis. The results of Lemma 4.1 is based on a sequence of lemmas introduced below.

**Lemma C.1.** *Suppose that the objective function $f$ satisfies Assumption 1. At each stage $t$, the step-size $\eta_i^t$ is monotonically decreasing. Then, for $i \ge 2$, we have*

$$\mathbb{E}[f(z_{i+1}^t) \mid \mathcal{F}_i^t] - f(z_i^t) + \frac{\beta}{1-\beta}\left(1 - \frac{\eta_i^t}{\eta_{i-1}^t}\right)\left(f(x_i^t) - f(x_{i-1}^t)\right)$$

$$\le -\eta_i^t \left\|\nabla f(x_i^t)\right\|^2 + \frac{\beta L}{2(1-\beta)^2}\left\|x_i^t - x_{i-1}^t\right\|^2 + \frac{L}{2(1-\beta)^2}\mathbb{E}[\left\|x_{i+1}^t - x_i^t\right\|^2 \mid \mathcal{F}_i^t].$$

*Proof.* Using the $L$-smoothness of $f$ (Assumption 1) and taking conditional expectation gives

$$\mathbb{E}[f(z_{i+1}^t) \mid \mathcal{F}_i^t] - f(x_i^t)$$

$$\leq \mathbb{E}[\langle \nabla f(x_i^t), z_{i+1}^t - x_i^t \rangle \mid \mathcal{F}_i^t] + \frac{L}{2}\mathbb{E}[\|z_{i+1}^t - x_i^t\|^2 \mid \mathcal{F}_i^t]$$

$$\leq \frac{1}{1-\beta}\mathbb{E}[\langle \nabla f(x_i^t), x_{i+1}^t - x_i^t \rangle \mid \mathcal{F}_i^t] + \frac{L}{2(1-\beta)^2}\mathbb{E}[\|x_{i+1}^t - x_i^t\|^2 \mid \mathcal{F}_i^t]$$

$$\leq \frac{1}{1-\beta}\mathbb{E}\langle \nabla f(x_i^t), -\eta_i^t((1-\beta)g_i^t + \beta v_i^t)\rangle \mid \mathcal{F}_i^t] + \frac{L}{2(1-\beta)^2}\mathbb{E}[\|x_{i+1}^t - x_i^t\|^2 \mid \mathcal{F}_i^t]$$

$$\leq -\eta_i^t \|\nabla f(x_i^t)\|^2 - \frac{\eta_i^t \beta}{1-\beta}\langle \nabla f(x_i^t), v_i^t\rangle + \frac{L}{2(1-\beta)^2}\mathbb{E}[\|x_{i+1}^t - x_i^t\|^2 \mid \mathcal{F}_i^t]$$

$$= -\eta_i^t \|\nabla f(x_i^t)\|^2 - \frac{\beta\eta_i^t}{(1-\beta)\eta_{i-1}^t}\langle \nabla f(x_i^t), x_{i-1}^t - x_i^t\rangle + \frac{L}{2(1-\beta)^2}\mathbb{E}[\|x_{i+1}^t - x_i^t\|^2 \mid \mathcal{F}_i^t].$$

Re-using the $L$-smoothness property of $f$ at $z_i^t$ and $x_i^t$ gives

$$f(z_i^t) \geq f(x_i^t) + \langle \nabla f(x_i^t), z_i^t - x_i^t\rangle - \frac{L}{2}\|z_i^t - x_i^t\|^2$$

$$= f(x_i^t) + \frac{\beta}{1-\beta}\langle \nabla f(x_i^t), x_i^t - x_{i-1}^t\rangle - \frac{L\beta^2}{2(1-\beta)^2}\|x_i^t - x_{i-1}^t\|^2. \tag{39}$$

Then, combining the two inequalities above, we have

$$\mathbb{E}[f(z_{i+1}^t) \mid \mathcal{F}_i^t] \leq f(z_i^t) - \eta_i^t \|\nabla f(x_i^t)\|^2 + \frac{\beta}{1-\beta}\left(1 - \frac{\eta_i^t}{\eta_{i-1}^t}\right)\langle \nabla f(x_i^t), x_{i-1}^t - x_i^t\rangle$$

$$+ \frac{L}{2(1-\beta)^2}\mathbb{E}[\|x_{i+1}^t - x_i^t\|^2 \mid \mathcal{F}_i^t] + \frac{L\beta^2}{2(1-\beta)^2}\|x_i^t - x_{i-1}^t\|^2. \tag{40}$$

The step-size $\eta_i^t$ for each stage is monotonically decreasing, i.e. $\eta_i^t \leq \eta_{i-1}^t$ for $i \geq 2$, so $\left(1 - \frac{\eta_i^t}{\eta_{i-1}^t}\right) \geq 0$. By the $L$-smoothness of $f$, the inner product of (40) can be estimated as

$$\langle \nabla f(x_i^t), x_{i-1}^t - x_i^t\rangle \leq f(x_{i-1}^t) - f(x_i^t) + \frac{L}{2}\|x_i^t - x_{i-1}^t\|^2. \tag{41}$$

Applying (41) into (40), we find

$$\mathbb{E}[f(z_{i+1}^t) \mid \mathcal{F}_i^t]$$

$$\leq f(z_i^t) - \eta_i^t \|\nabla f(x_i^t)\|^2 + \frac{\beta}{1-\beta}\left(1 - \frac{\eta_i^t}{\eta_{i-1}^t}\right)\left(f(x_{i-1}^t) - f(x_i^t) + \frac{L}{2}\|x_i^t - x_{i-1}^t\|^2\right)$$

$$+ \frac{L}{2(1-\beta)^2}\mathbb{E}[\|x_{i+1}^t - x_i^t\|^2 \mid \mathcal{F}_i^t] + \frac{L\beta^2}{2(1-\beta)^2}\|x_i^t - x_{i-1}^t\|^2. \tag{42}$$

Finally, we re-write the above inequality as

$$\mathbb{E}[f(z_{i+1}^t) \mid \mathcal{F}_i^t] - f(z_i^t) + \frac{\beta}{1-\beta}\left(1 - \frac{\eta_i^t}{\eta_{i-1}^t}\right)(f(x_i^t) - f(x_{i-1}^t))$$

$$\leq -\eta_i^t \|\nabla f(x_i^t)\|^2 + \left(\frac{\beta^2 L}{2(1-\beta)^2} + \frac{\beta L}{2(1-\beta)}\right)\|x_i^t - x_{i-1}^t\|^2 + \frac{L}{2(1-\beta)^2}\mathbb{E}[\|x_{i+1}^t - x_i^t\|^2 \mid \mathcal{F}_i^t]$$

$$\leq -\eta_i^t \|\nabla f(x_i^t)\|^2 + \frac{\beta L}{2(1-\beta)^2}\|x_i^t - x_{i-1}^t\|^2 + \frac{L}{2(1-\beta)^2}\mathbb{E}[\|x_{i+1}^t - x_i^t\|^2 \mid \mathcal{F}_i^t].$$

The proof is complete. $\qquad\square$

**Lemma C.2.** *Suppose that the objective function satisfies Assumption 1 and the step-size $\eta_i^t$ is monotonically decreasing with $\eta_i^t \leq \frac{1}{L}$ at each stage, then*

$$\mathbb{E}\left[\|x_{i+1}^t - x_i^t\|^2 \mid \mathcal{F}_i^t\right] - \beta^2\|x_i^t - x_{i-1}^t\|^2 + 2(1-\beta)\eta_i^t\left(\mathbb{E}[f(x_{i+1}^t) \mid \mathcal{F}_i^t] - f(x_i^t)\right)$$

$$\leq -2(\eta_i^t)^2(1-\beta)^2\|\nabla f(x_i^t)\|^2 + 2(\eta_i^t)^2(1-\beta)^2\mathbb{E}\left[\|g_i^t\|^2 \mid \mathcal{F}_i^t\right] + (\eta_i^t)^3\beta(1-\beta)L\|v_i^t\|^2.$$

*Proof.* First, due to the $L$-smoothness of the objective function $f$, we have

$$f(x_{i+1}^t) \le f(x_i^t) + \langle \nabla f(x_i^t), x_{i+1}^t - x_i^t \rangle + \frac{L}{2} \left\| x_{i+1}^t - x_i^t \right\|^2$$

$$\le f(x_i^t) + \langle \nabla f(x_i^t), -\eta_i^t((1-\beta)g_i^t + \beta v_i^t) \rangle + \frac{(\eta_i^t)^2 L}{2} \left\| (1-\beta)g_i^t + \beta v_i^t \right\|^2$$

$$\overset{(a)}{\le} f(x_i^t) - (1-\beta)\eta_i^t \langle \nabla f(x_i^t), g_i^t \rangle - \beta\eta_i^t \langle \nabla f(x_i^t), v_i^t \rangle + \frac{(\eta_i^t)^2 L}{2} \left( (1-\beta) \left\| g_i^t \right\|^2 + \beta \left\| v_i^t \right\|^2 \right)$$

where inequality (a) follows the *Cauchy-Schwarz* inequality that $\left\| (1-\beta)g_i^t + \beta v_i^t \right\|^2 \le (1-\beta) \left\| g_i^t \right\|^2 + \beta \left\| v_i^t \right\|^2$. Then taking conditional expectation on both sides and due to that $g_i^t$ is an unbiased estimator of $\nabla f(x_i^t)$, i.e., $\mathbb{E}[g_i^t \mid \mathcal{F}_i^t] = \nabla f(x_i^t)$, we have

$$\mathbb{E}[f(x_{i+1}^t) \mid \mathcal{F}_i^t] \le f(x_i^t) - \eta_i^t(1-\beta) \left\| \nabla f(x_i^t) \right\|^2 - \beta\eta_i^t \langle \nabla f(x_i^t), v_i^t \rangle$$
$$+ \frac{(\eta_i^t)^2(1-\beta)L}{2} \mathbb{E}[\left\| g_i^t \right\|^2 \mid \mathcal{F}_i^t] + \frac{(\eta_i^t)^2 \beta L}{2} \left\| v_i^t \right\|^2. \tag{43}$$

We recall the definition of $v_{i+1}^t$ and incorporate (35) into (36), then

$$\mathbb{E}[\left\| x_{i+1}^t - x_i^t \right\|^2 \mid \mathcal{F}_i^t]$$

$$= \mathbb{E}[\left\| \eta_i^t(\beta v_i^t + (1-\beta)g_i^t) \right\|^2 \mid \mathcal{F}_i^t] = (\eta_i^t)^2 \mathbb{E}[\left\| \beta v_i^t + (1-\beta)g_i^t \right\|^2 \mid \mathcal{F}_i^t]$$

$$\overset{(a)}{=} (\eta_i^t)^2 \left( \beta^2 \left\| v_i^t \right\|^2 + (1-\beta)^2 \mathbb{E}[\left\| g_i^t \right\|^2 \mid \mathcal{F}_i^t] + 2\beta(1-\beta) \langle v_i^t, \nabla f(x_i^t) \rangle \right)$$

$$\overset{(b)}{=} \left( \frac{\eta_i^t}{\eta_{i-1}^t} \right)^2 \beta^2 \left\| x_i^t - x_{i-1}^t \right\|^2 + (\eta_i^t)^2(1-\beta) \left( (1-\beta)\mathbb{E}[\left\| g_i^t \right\|^2] + 2\beta \langle v_i^t, \nabla f(x_i^t) \rangle \right) \tag{44}$$

$$\overset{(c)}{\le} \beta^2 \left\| x_i^t - x_{i-1}^t \right\|^2 + (\eta_i^t)^2(1-\beta) \left( (1-\beta)\mathbb{E}[\left\| g_i^t \right\|^2] + 2\beta \langle v_i^t, \nabla f(x_i^t) \rangle \right), \tag{45}$$

where $(a)$ uses the fact that $\mathbb{E}[g_i^t \mid \mathcal{F}_i^t] = \nabla f(x_i^t)$; $(b)$ follows the procedure that $x_i^t = x_{i-1}^t - \eta_{i-1}^t v_i^t$; (c) applies the fact that the step-size per stage is monotonically decreasing, i.e., $\eta_i^t \le \eta_{i-1}^t$ for $i \ge 2$. Then multiplying $2\eta_i^t(1-\beta)$ into (43) and combining (45), we get that

$$\mathbb{E}\left[ \left\| x_{i+1}^t - x_i^t \right\|^2 \mid \mathcal{F}_i^t \right] - \beta^2 \left\| x_i^t - x_{i-1}^t \right\|^2 + 2\eta_i^t(1-\beta) \left( \mathbb{E}[f(x_{i+1}^t) \mid \mathcal{F}_i^t] - f(x_i^t) \right)$$

$$\le -2(\eta_i^t)^2(1-\beta)^2 \left\| \nabla f(x_i^t) \right\|^2$$

$$+ (\eta_i^t)^2(1-\beta)^2(L\eta_i^t + 1)\mathbb{E}\left[ \left\| g_i^t \right\|^2 \mid \mathcal{F}_i^t \right] + (\eta_i^t)^3 \beta(1-\beta)L \left\| v_i^t \right\|^2$$

$$\le -2(\eta_i^t)^2(1-\beta)^2 \left\| \nabla f(x_i^t) \right\|^2 + 2(\eta_i^t)^2(1-\beta)^2 \mathbb{E}\left[ \left\| g_i^t \right\|^2 \mid \mathcal{F}_i^t \right] + (\eta_i^t)^3 \beta(1-\beta)L \left\| v_i^t \right\|^2$$

where the last inequality follows from the fact that $\eta_i^t \le 1/L$. $\qquad\square$

*Proof.* (**of Lemma 4.1**) First we apply the result of Lemma C.1 and divided by $\eta_i^t$ to the both side, we have

$$\frac{\mathbb{E}[f(z_{i+1}^t) \mid \mathcal{F}_i^t] - f(z_i^t)}{\eta_i^t} + \frac{\beta}{1-\beta} \frac{\left( f(x_i^t) - f(x_{i-1}^t) \right)}{\eta_i^t} - \frac{\beta}{1-\beta} \frac{\left( f(x_i^t) - f(x_{i-1}^t) \right)}{\eta_{i-1}^t}$$

$$\le - \left\| \nabla f(x_i^t) \right\|^2 + \left( \frac{\beta L}{2(1-\beta)^2 \eta_i^t} \right) \left\| x_i^t - x_{i-1}^t \right\|^2 + \frac{\eta_i^t L}{2(1-\beta)^2} \mathbb{E}[\left\| v_{i+1}^t \right\|^2 \mid \mathcal{F}_i^t]. \tag{46}$$

Then we recall the result of Lemma C.2

$$\mathbb{E}[\left\| x_{i+1}^t - x_i^t \right\|^2 \mid \mathcal{F}_i^t] - \left\| x_i^t - x_{i-1}^t \right\|^2 + 2\eta_i^t \left( \mathbb{E}[f(x_{i+1}^t)] - f(x_i^t) \right)$$

$$\le -(1-\beta^2) \left\| x_i^t - x_{i-1}^t \right\|^2 - 2(\eta_i^t)^2(1-\beta)^2 \left\| \nabla f(x_i^t) \right\|^2$$

$$+ 2(\eta_i^t)^2(1-\beta)^2 \mathbb{E}\left[ \left\| g_i^t \right\|^2 \mid \mathcal{F}_i^t \right] + (\eta_i^t)^3 \beta(1-\beta)L \left\| v_i^t \right\|^2,$$

multiplying a constant $r = \frac{\beta L}{2(1-\beta^2)(1-\beta)^2} > 0$ and dividing $\eta_i^t$ to the both side, and then incorporating it into (46), we have

$$
\begin{aligned}
&\frac{\mathbb{E}[f(z_{i+1}^t) \mid \mathcal{F}_i^t] - f(z_i^t)}{\eta_i^t} + \frac{\beta}{1-\beta} \frac{\left(f(x_i^t) - f(x_{i-1}^t)\right)}{\eta_i^t} - \frac{\beta}{1-\beta} \frac{\left(f(x_i^t) - f(x_{i-1}^t)\right)}{\eta_{i-1}^t} \\
&+ \frac{r}{\eta_i^t}\left(\mathbb{E}[\|x_{i+1}^t - x_i^t\|^2 \mid \mathcal{F}_i^t] - \|x_i^t - x_{i-1}^t\|^2\right) + 2r\left(\mathbb{E}[f(x_{i+1}^t) \mid \mathcal{F}_i^t] - f(x_i^t)\right) \\
&\leq -\|\nabla f(x_i^t)\|^2 + 2r(\eta_i^t)(1-\beta)^2 \mathbb{E}[\|g_i^t\|^2 \mid \mathcal{F}_i^t] + r(\eta_i^t)^2\beta(1-\beta)L \|v_i^t\|^2 \\
&+ \frac{\eta_i^t L}{2(1-\beta)^2} \mathbb{E}[\|v_{i+1}^t\|^2 \mid \mathcal{F}_i^t].
\end{aligned}
\tag{47}
$$

We define a function $W_{i+1}^t$ as follows:

$$
W_{i+1}^t = \frac{f(z_{i+1}^t) - f^*}{\eta_i^t} + \frac{r\|x_{i+1}^t - x_i^t\|^2}{\eta_i^t} + 2r[f(x_{i+1}^t) - f^*].
$$

Because of $\eta_i^t \leq \eta_{i-1}^t$ at each stage, we have $-1/\eta_i^t \leq -1/\eta_{i-1}^t$ ($i \geq 2$), then

$$
\begin{aligned}
W_{i+1}^t \leq{}& \frac{f(z_{i+1}^t) - f^*}{\eta_i^t} + \frac{\beta}{1-\beta} \frac{(f(x_i^t) - f^*)}{\eta_i^t} - \frac{\beta}{1-\beta} \frac{(f(x_i^t) - f^*)}{\eta_{i-1}^t} \\
&+ \frac{r}{\eta_i^t}\|x_{i+1}^t - x_i^t\|^2 + 2r\left(f(x_{i+1}^t) - f^*\right).
\end{aligned}
\tag{48}
$$

Taking conditional expectation on $W_{i+1}^t$ and applying (47) to the above inequality, we have

$$
\begin{aligned}
\mathbb{E}[W_{i+1}^t \mid \mathcal{F}_i^t] \leq{}& W_i^t + (f(z_i^t) - f^*)\left(\frac{1}{\eta_i^t} - \frac{1}{\eta_{i-1}^t}\right) + \frac{\beta\left(f(x_{i-1}^t) - f^*\right)}{1-\beta}\left(\frac{1}{\eta_i^t} - \frac{1}{\eta_{i-1}^t}\right) \\
&+ r\|x_i^t - x_{i-1}^t\|^2\left(\frac{1}{\eta_i^t} - \frac{1}{\eta_{i-1}^t}\right) - \|\nabla f(x_i^t)\|^2 + 2r(\eta_i^t)(1-\beta)^2\mathbb{E}[\|g_i^t\|^2 \mid \mathcal{F}_i^t] \\
&+ r(\eta_i^t)^2\beta(1-\beta)L\|v_i^t\|^2 + \frac{\eta_i^t L}{2(1-\beta)^2}\mathbb{E}[\|v_{i+1}^t\|^2 \mid \mathcal{F}_i^t].
\end{aligned}
\tag{49}
$$

We recall that $v_1^1 = 0$, due to the assumption that $\mathbb{E}[\|g_i^t\|^2] \leq G^2$, and $v_{i+1}^t$ is a convex combination of $g_i^t$ and $v_i^t$, then by induction if $\mathbb{E}[\|v_i^t\|^2] \leq G^2$, then

$$
\mathbb{E}[\|v_{i+1}^t\|^2] = \mathbb{E}\|\beta v_i^t + (1-\beta)g_i^t\|^2 \leq \beta\|v_i^t\|^2 + (1-\beta)\mathbb{E}[\|g_i^t\|^2] \leq G^2.
\tag{50}
$$

Therefore, we have $\mathbb{E}[\|v_i^t\|^2]$ is bounded by $G^2$. Then we apply $\mathbb{E}[\|g_i^t\|^2] \leq G^2$, $\mathbb{E}[\|v_i^t\|^2] \leq G^2$ and $\eta_i^t \leq 1/L$ into (49)

$$
\begin{aligned}
\mathbb{E}[W_{i+1}^t \mid \mathcal{F}_i^t] - W_i^t \leq{}& (f(z_i^t) - f^*)\left(\frac{1}{\eta_i^t} - \frac{1}{\eta_{i-1}^t}\right) + \frac{\beta\left(f(x_{i-1}^t) - f^*\right)}{1-\beta}\left(\frac{1}{\eta_i^t} - \frac{1}{\eta_{i-1}^t}\right) \\
&+ r\|x_i^t - x_{i-1}^t\|^2\left(\frac{1}{\eta_i^t} - \frac{1}{\eta_{i-1}^t}\right) - \|\nabla f(x_i^t)\|^2 \\
&+ \eta_i^t G^2\left(r(1-\beta)(2-\beta) + \frac{L}{2(1-\beta)^2}\right).
\end{aligned}
\tag{51}
$$

The step-size is decreasing at each stage, then $\eta_i^t \leq \eta_{i-1}^t$ ($i \geq 2$), thus $\frac{1}{\eta_i^t} - \frac{1}{\eta_{i-1}^t} \geq 0$. Due to the fact that $v_i^t$ is bounded (see (50)), i.e., $\mathbb{E}[\|v_i^t\|^2] \leq G^2$, we have

$$
\mathbb{E}[\|x_i^t - x_{i-1}^t\|^2] = (\eta_{i-1}^t)^2 \mathbb{E}[\|v_i^t\|^2]] \leq (\eta_{i-1}^t)^2 G^2 \leq \frac{G^2}{L^2}.
\tag{52}
$$

Recalling the definition of $z_i^t$, and applying the assumption that $\mathbb{E}[f(x_i^t) - f^*] \le \Delta_0$ for each $t, i \ge 1$ and $f$ is $L$-smooth on its domain, and $\eta_i^t \le 1/L$ gives

$$
\begin{aligned}
f(z_i^t) &\le f(x_i^t) + \langle \nabla f(x_i^t), z_i^t - x_i^t \rangle + \frac{L}{2} \left\| z_i^t - x_i^t \right\|^2 \\
&\le f(x_i^t) + \frac{\beta}{1-\beta} \langle \nabla f(x_i^t), x_i^t - x_{i-1}^t \rangle + \frac{L\beta^2}{2(1-\beta)^2} \left\| x_i^t - x_{i-1}^t \right\|^2 \\
&\overset{(a)}{\le} f(x_i^t) + \frac{\beta}{1-\beta} \left( f(x_i^t) - f(x_{i-1}^t) + \frac{L}{2} \left\| x_i^t - x_{i-1}^t \right\|^2 \right) + \frac{L\beta^2}{2(1-\beta)^2} \left\| x_i^t - x_{i-1}^t \right\|^2 \\
&\le f(x_i^t) + \frac{\beta}{1-\beta} \left( f(x_i^t) - f(x_{i-1}^t) \right) + \frac{L\beta}{2(1-\beta)^2} \left\| x_i^t - x_{i-1}^t \right\|^2 \\
&\le \frac{1}{1-\beta} f(x_i^t) - \frac{\beta}{1-\beta} f(x_{i-1}^t) + \frac{L\beta}{2(1-\beta)^2} \left\| x_i^t - x_{i-1}^t \right\|^2
\end{aligned}
\tag{53}
$$

where the inequality (a) dues to the fact that $f(x_{i-1}^t) \le f(x_i^t) + \langle \nabla f(x_i^t), x_{i-1}^t - x_i^t \rangle + \frac{L}{2} \left\| x_{i-1}^t - x_i^t \right\|^2$. Then we have

$$
\mathbb{E}[f(z_i^t) - f^*] \le \frac{1}{1-\beta} \Delta_0 + \frac{L\beta}{2(1-\beta)^2} (\eta_{i-1}^t)^2 G^2 \le \frac{\Delta_0}{1-\beta} + \frac{\beta G^2}{2(1-\beta)^2 L}.
\tag{54}
$$

Let $\Delta_z = \frac{\Delta_0}{1-\beta} + \frac{\beta G^2}{2(1-\beta)^2 L}$. Finally, applying (52) and (54), the bounded assumption on $\mathbb{E}[f(x_i^t) - f^*]$, and $\eta_i^t \le 1/L$, we have

$$
\begin{aligned}
\mathbb{E}[W_{i+1}^t \mid \mathcal{F}_i^t] &\le W_i^t + \left( \frac{\beta \Delta_0}{1-\beta} + \Delta_z + \frac{r G^2}{L^2} \right) \left( \frac{1}{\eta_i^t} - \frac{1}{\eta_{i-1}^t} \right) - \left\| \nabla f(x_i^t) \right\|^2 \\
&\quad + \eta_i^t \left( r(1-\beta)(2-\beta) + \frac{L}{2(1-\beta)^2} \right) G^2 \\
&= W_i^t + A_1 \left( \frac{1}{\eta_i^t} - \frac{1}{\eta_{i-1}^t} \right) - \left\| \nabla f(x_i^t) \right\|^2 + \eta_i^t B_1 G^2
\end{aligned}
\tag{55}
$$

where $A_1 = \frac{\beta \Delta_0}{1-\beta} + \Delta_z + \frac{r G^2}{L^2}$, $B_1 = r(1-\beta)(2-\beta) + \frac{L}{2(1-\beta)^2}$ and $\Delta_z = \frac{\Delta_0}{1-\beta} + \frac{\beta G^2}{2(1-\beta)^2 L}$. $\quad\square$

The bandwidth step-size highly rises the difficulty of the analysis for momentum, especially when the step-size has an increase between the stages, i.e. $\eta_{S_{t-1}}^{t-1} := \eta_0^t < \eta_1^t$. Before giving the results, we consider two situations:

- $\eta_0^t > \eta_1^t$. We can apply Lemma 4.1 from $i = 1$ to $S_t$. Recalling the definition of $W_{i+1}^t$, we have $W_1^{t+1} = W_{S_t+1}^t$, then

$$
\sum_{i=1}^{S_t} \mathbb{E}[\left\| \nabla f(x_i^t) \right\|^2] \le \left( \mathbb{E}[W_1^t] - \mathbb{E}[W_1^{t+1}] \right) + A_1 \left( \frac{1}{\eta_{S_t}^t} - \frac{1}{\eta_0^t} \right) + B_1 G^2 \sum_{i=1}^{S_t} \eta_i^t.
\tag{56}
$$

- Otherwise if $\eta_0^t \le \eta_1^t$, the results of Lemma 4.1 only hold from $i = 2$ to $S_t$. Then

$$
\begin{aligned}
\sum_{i=1}^{S_t} \mathbb{E}[\left\| \nabla f(x_i^t) \right\|^2] &\le \left( \mathbb{E}[W_2^t] - \mathbb{E}[W_1^t] + \mathbb{E}[W_1^t] - \mathbb{E}[W_1^{t+1}] \right) + \mathbb{E}[\left\| \nabla f(x_1^t) \right\|^2] \\
&\quad + A_1 \left( \frac{1}{\eta_{S_t}^t} - \frac{1}{\eta_1^t} \right) + B_1 G^2 \sum_{i=2}^{S_t} \eta_i^t.
\end{aligned}
\tag{57}
$$

For the bandwidth step-size, the initial step-size of stage $t$, $\eta_1^t$, is possibly larger than the ending step-size of the previous stage, $\eta_0^t$. Thus, we can not use the simpler condition (56), but have to rely on (57) in our derivations below.

**Lemma C.3.** *Suppose the same setting as Lemma 4.1, we have*

$$\mathbb{E}[\|\nabla f(\hat{x}_T)\|^2] \leq \frac{1}{\sum_\delta}\left(W_1^1 + \frac{C_0}{\delta(N+1)} + \frac{\Delta_z}{m\delta(N)\delta(N+1)} + \frac{C_1}{m}\sum_{t=1}^{N}\frac{1}{\delta(t)^2} + C_2\sum_{t=1}^{N}\frac{1}{\delta(t)} + B_1 G^2 MT\right)$$

*where $\sum_\delta = \sum_{t=1}^{N} S_t \delta(t)^{-1}$, $C_0 = r(\frac{G^2}{L} + 2\Delta_0)$, $C_1 = A_1 + \Delta_z + \frac{\Delta_0}{1-\beta}$, $C_2 = C_0 + A_2 G^2$, and $A_1$, $B_1$, $r$, and $\Delta_z$ are defined in Lemma 4.1.*

*Proof.* Applying the result of Lemma 4.1 from $i = 2$ to $S_t$, the step-size $\eta_i^t \in [m\delta(t), M\delta(t)]$, and $W_1^{t+1} = W_{S_t+1}^t$, we have

$$\sum_{i=1}^{S_t}\mathbb{E}[\|\nabla f(x_i^t)\|^2] \leq \left(\mathbb{E}[W_2^t] - \mathbb{E}[W_1^t] + \mathbb{E}[W_1^t] - \mathbb{E}[W_1^{t+1}]\right) + \mathbb{E}[\|\nabla f(x_1^t)\|^2]$$

$$+ A_1\left(\frac{1}{\eta_{S_t}^t} - \frac{1}{\eta_1^t}\right) + B_1 G^2 M(S_t - 1)\delta(t). \tag{58}$$

Recalling the output of Algorithm 2, we have

$$\mathbb{E}[\|\nabla f(\hat{x}_T)\|^2] = \frac{1}{\sum_{t=1}^{N} S_t \delta(t)^{-1}}\sum_{t=1}^{N}\delta(t)^{-1}\sum_{i=1}^{S_t}\mathbb{E}[\|\nabla f(x_i^t)\|^2]. \tag{59}$$

Then we divide $\delta(t)$ into the both side of (58), apply (58) from $t = 1$ to $N$ and let $\sum_\delta = \sum_{t=1}^{N} S_t \delta(t)^{-1}$

$$\mathbb{E}[\|\nabla f(\hat{x}_T)\|^2] \leq \frac{1}{\sum_\delta}\left(\sum_{t=1}^{N}\frac{\mathbb{E}[W_2^t] - \mathbb{E}[W_1^t] + \mathbb{E}[W_1^t] - \mathbb{E}[W_1^{t+1}]}{\delta(t)} + A_1\sum_{t=1}^{N}\frac{1}{\delta(t)}\left(\frac{1}{\eta_{S_t}^t} - \frac{1}{\eta_0^t}\right)\right)$$

$$+ \frac{1}{\sum_\delta}\left(\sum_{t=1}^{N}\frac{\mathbb{E}[\|\nabla f(x_1^t)\|^2]}{\delta(t)} + B_1 G^2 MT\right). \tag{60}$$

First, we estimate $\sum_{t=1}^{N}\frac{\mathbb{E}[\|\nabla f(x_1^t)\|^2]}{\delta(t)}$. From Lemma C.2, let $i = 1$, then incorporating the inequalities (43) and (44), we have

$$2(\eta_1^t)^2(1-\beta)^2\mathbb{E}[\|\nabla f(x_1^t)\|^2] \leq \left(\frac{\eta_1^t \beta}{\eta_0^t}\right)^2\|x_1^t - x_0^t\|^2 - \mathbb{E}[\|x_2^t - x_1^t\|^2] + 2(\eta_1^t(1-\beta))^2\mathbb{E}[\|g_1^t\|^2]$$

$$+ (\eta_1^t)^3\beta(1-\beta)L\|v_1^t\|^2 - 2\eta_1^t(1-\beta)\left(f(x_1^t) - \mathbb{E}[f(x_2^t)]\right).$$

Then dividing $2(\eta_1^t)^2(1-\beta)^2$ to the both side and applying the fact that $\|x_1^t - x_0^t\| = (\eta_0^t)^2\|v_1^t\|^2$, $\mathbb{E}[\|g_i^t\|^2] \leq G^2$ and $\mathbb{E}[\|v_i^t\|^2] \leq G^2$ for any $i, t$, $f(x_2^t) - f^* \leq \Delta_0$ and $\eta_1^t \geq m\delta(t)$, we have

$$\mathbb{E}[\|\nabla f(x_1^t)\|^2] \leq G^2\left(1 + \frac{\beta}{2(1-\beta)^2}\right) + \frac{\Delta_0}{(1-\beta)\eta_1^t} \leq A_2 G^2 + \frac{\Delta_0}{(1-\beta)m\delta(t)} \tag{61}$$

where $A_2 = \left(1 + \frac{\beta}{2(1-\beta)^2}\right)$. Then

$$\sum_{t=1}^{N}\frac{\mathbb{E}[\|\nabla f(x_1^t)\|^2]}{\delta(t)} \leq \sum_{t=1}^{N}\frac{A_2 G^2}{\delta(t)} + \frac{\Delta_0}{(1-\beta)m}\sum_{t=1}^{N}\frac{1}{\delta^2(t)}. \tag{62}$$

Next we turn to estimate $\sum_{t=1}^{N}\frac{\mathbb{E}[W_2^t - W_1^t]}{\delta(t)}$. Recalling the definition of $W_{i+1}^t$, we have $W_i^t \geq 0$. Applying the inequalities (52), (54) and the assumption that $f(x_i^t) - f^* \leq \Delta_0$ for any $i, t$, we have

$$\mathbb{E}[W_2^t] - \mathbb{E}[W_1^t] \leq \mathbb{E}[W_2^t] := \frac{\mathbb{E}[f(z_2^t) - f^*]}{\eta_1^t} + \frac{r\mathbb{E}[\|x_2^t - x_1^t\|^2]}{\eta_1^t} + 2r\mathbb{E}[f(x_2^t) - f^*]$$

$$\leq \frac{\Delta_z}{m\delta(t)} + r\left(\frac{G^2}{L} + 2\Delta_0\right), \tag{63}$$

dividing $\delta(t)$ and applying (63) from $t = 1$ to $N$, we have

$$\sum_{t=1}^{N} \frac{\mathbb{E}[W_2^t] - \mathbb{E}[W_1^t]}{\delta(t)} \leq \frac{\Delta_z}{m} \sum_{t=1}^{N} \frac{1}{\delta(t)^2} + r\left(\frac{G^2}{L} + 2\Delta_0\right) \sum_{t=1}^{N} \frac{1}{\delta(t)}. \tag{64}$$

Then we consider

$$\sum_{t=1}^{N} \frac{\mathbb{E}[W_1^t] - \mathbb{E}[W_1^{t+1}]}{\delta(t)} = \sum_{t=1}^{N} \left(\frac{\mathbb{E}[W_1^t]}{\delta(t)} - \frac{\mathbb{E}[W_1^{t+1}]}{\delta_1(t+1)}\right) + \sum_{t=1}^{N} \left(\frac{1}{\delta_1(t+1)} - \frac{1}{\delta(t)}\right) \mathbb{E}[W_1^{t+1}]$$

$$\leq \frac{W_1^1}{\delta_1(1)} + \sum_{t=1}^{N} \left(\frac{1}{\delta_1(t+1)} - \frac{1}{\delta(t)}\right) \mathbb{E}[W_1^{t+1}]. \tag{65}$$

Recalling the definition of $\mathbb{E}[W_1^{t+1}]$,

$$\mathbb{E}[W_1^{t+1}] = \frac{\mathbb{E}[f(z_1^{t+1}) - f^*]}{\eta_{S_t}^t} + \frac{r\mathbb{E}[\|x_1^{t+1} - x_0^{t+1}\|^2]}{\eta_{S_t}^t} + 2r\mathbb{E}[f(x_1^{t+1}) - f^*],$$

and applying the assumption that $\mathbb{E}[f(x_i^t) - f^*] \leq \Delta_0$ and $\mathbb{E}[f(z_i^t) - f^*] \leq \Delta_z$, and $\eta_0^{t+1} = \eta_{S_t}^t$, $\mathbb{E}[\|x_1^{t+1} - x_0^{t+1}\|^2] = (\eta_{S_t}^t)^2 \mathbb{E}[\|v_1^{t+1}\|^2] \leq (\eta_{S_t}^t)^2 G^2$, we have

$$\mathbb{E}[W_1^{t+1}] \leq \frac{\Delta_z}{\eta_{S_t}^t} + r\eta_{S_t}^t G^2 + 2r\Delta_0 \overset{(a)}{\leq} \frac{\Delta_z}{m\delta(t)} + r\left(\frac{G^2}{L} + 2\Delta_0\right) \tag{66}$$

where (a) follows from $\eta_{S_t}^t \geq m\delta(t)$ and $\eta_{S_t}^t \leq 1/L$. Applying (66) into (65), we have

$$\sum_{t=1}^{N} \frac{\mathbb{E}[W_1^t] - \mathbb{E}[W_1^{t+1}]}{\delta(t)} \leq \frac{W_1^1}{\delta(1)} + \frac{r\left(\frac{G^2}{L} + 2\Delta_0\right)}{\delta(N+1)} + \frac{\Delta_z}{m} \sum_{t=1}^{N} \left(\frac{1}{\delta(t+1)\delta(t)} - \frac{1}{\delta(t)^2}\right)$$

$$\leq \frac{W_1^1}{\delta(1)} + \frac{r\left(\frac{G^2}{L} + 2\Delta_0\right)}{\delta(N+1)} + \frac{\Delta_z}{m} \sum_{t=1}^{N} \left(\frac{1}{\delta(t+1)\delta(t)} - \frac{1}{\delta(t)\delta(t-1)}\right)$$

$$\leq \frac{W_1^1}{\delta(1)} + \frac{r\left(\frac{G^2}{L} + 2\Delta_0\right)}{\delta(N+1)} + \frac{\Delta_z}{m\delta(N)\delta(N+1)}, \tag{67}$$

where the second inequality follows that $\delta(t)$ is decreasing, so $\delta(t) \leq \delta(t-1)$, then $-1/\delta(t) \leq -1/\delta(t-1)$. Finally, due to that $\eta_{S_t}^t \in [m\delta(t), M\delta(t)]$, we have

$$\sum_{t=1}^{N} \frac{1}{\delta(t)} \left(\frac{1}{\eta_{S_t}^t} - \frac{1}{\eta_0^t}\right) \leq \sum_{t=1}^{N} \frac{1}{m\delta(t)^2}. \tag{68}$$

Incorporate the inequalities (62), (64), (67) and (68) into (60), we have

$$\mathbb{E}[\|\nabla f(\hat{x}_T)\|^2] \leq \frac{1}{\sum_\delta} \left(\frac{W_1^1}{\delta(1)} + \frac{r\left(\frac{G^2}{L} + 2\Delta_0\right)}{\delta(N+1)} + \frac{\Delta_z}{m\delta(N)\delta(N+1)} + A_1 \sum_{t=1}^{N} \frac{1}{m\delta(t)^2}\right)$$

$$+ \frac{1}{\sum_\delta} \left(\frac{\Delta_z}{m} \sum_{t=1}^{N} \frac{1}{\delta(t)^2} + r\left(\frac{G^2}{L} + 2\Delta_0\right) \sum_{t=1}^{N} \frac{1}{\delta(t)} + B_1 G^2 M T\right)$$

$$+ \frac{1}{\sum_\delta} \left(\sum_{t=1}^{N} \frac{A_2 G^2}{\delta(t)} + \frac{\Delta_0}{(1-\beta)m} \sum_{t=1}^{N} \frac{1}{\delta_1^2(t)}\right). \tag{69}$$

The above result can be re-written as (recall $\delta(1) = 1$)

$$\mathbb{E}[\|\nabla f(\hat{x}_T)\|^2]$$

$$\leq \frac{1}{\sum_\delta} \left( W_1^1 + \frac{r\left(\frac{G^2}{L} + 2\Delta_0\right)}{\delta(N+1)} + \frac{\Delta_z}{m\delta(N)\delta(N+1)} + \frac{(A_1 + \Delta_z + \frac{\Delta_0}{1-\beta})}{m} \sum_{t=1}^{N} \frac{1}{\delta(t)^2} \right)$$

$$+ \frac{1}{\sum_\delta} \left( \left( r\left(\frac{G^2}{L} + 2\Delta_0\right) + A_2 G^2 \right) \sum_{t=1}^{N} \frac{1}{\delta(t)} + B_1 G^2 M T \right)$$

$$\leq \frac{1}{\sum_\delta} \left( W_1^1 + \frac{C_0}{\delta(N+1)} + + \frac{\Delta_z}{m\delta(N)\delta(N+1)} + \frac{C_1}{m} \sum_{t=1}^{N} \frac{1}{\delta(t)^2} + C_2 \sum_{t=1}^{N} \frac{1}{\delta(t)} + B_1 G^2 M T \right)$$

where $C_0 = r(\frac{G^2}{L} + 2\Delta_0)$, $C_1 = A_1 + \Delta_z + \frac{\Delta_0}{1-\beta}$, and $C_2 = C_0 + A_2 G^2$. $\qquad\square$

*Proof.* (**of Theorem 4.2**) In this case, given the total number iteration $T \geq 1$, the number of stages $N \geq 1$, $S_t = S = \lceil T/N \rceil$, $\delta(t) = 1/\alpha^{t-1}$ for each $1 \leq t \leq N$, then the boundary function at the final stage $\delta(N) = 1/\alpha^{N-1}$ and $\delta(N+1) = 1/\alpha^N$. Applying the specific value of $\delta(t)$ and $N$ gives

$$\sum_{t=1}^{N} \frac{1}{\delta(t)} = \frac{\alpha^N - 1}{\alpha - 1} \tag{70}$$

$$\sum_{t=1}^{N} \frac{1}{\delta(t)^2} = \frac{\alpha^{2N} - 1}{\alpha^2 - 1}. \tag{71}$$

Then by $S_t = \lceil T/N \rceil$ and (70), we easily get

$$\sum_\delta = \sum_{t=1}^{N} S_t/\delta(t) \geq \frac{T(\alpha^N - 1)}{N(\alpha - 1)}. \tag{72}$$

We then plug the above results into Lemma C.3,

$$\mathbb{E}[\|\nabla f(\hat{x}_T)\|^2]$$

$$\leq \frac{N(\alpha - 1)}{T(\alpha^N - 1)} \left( W_1^1 + C_0 \alpha^N + \frac{\Delta_z \alpha^{2N-1}}{m} + \frac{C_1(\alpha^{2N} - 1)}{m(\alpha^2 - 1)} + \frac{C_2(\alpha^N - 1)}{\alpha - 1} + M B_1 G^2 T \right)$$

$$\leq W_1^1 \frac{N}{T\alpha^{N-1}} + (\alpha C_0 + C_2) \frac{N}{T} + \frac{(\Delta_z + C_1)}{m} \frac{N\alpha^N}{T} + M B_1 G^2 \frac{N}{\alpha^{N-1}}. \tag{73}$$

where $C_0 = r(\frac{G^2}{L} + 2\Delta_0)$, $C_1 = A_1 + \Delta_z + \frac{\Delta_0}{1-\beta}$, and $C_2 = C_0 + A_2 G^2$, $A_2 = 1 + \frac{\beta}{2(1-\beta)^2}$, and $W_1^1, A_1, B_1, \Delta_z$, and $r$ are defined in Lemma 4.1.

Especially, we consider the number of outer-stage $N = \lfloor (\log_\alpha T)/2 \rfloor$, the stage length $S_t = \lceil 2T/\log_\alpha T \rceil$, and the boundary functions $\delta(t) = 1/\alpha^{t-1}$ for all $t \in \{1, 2, \cdots, N\}$. Let $N = \lfloor (\log_\alpha T)/2 \rfloor$, we have

$$\mathbb{E}[\|\nabla f(\hat{x}_T)\|^2] \leq \frac{\alpha^2 W_1^1}{2\ln\alpha} \frac{\ln T}{T^{3/2}} + \frac{(\alpha C_0 + C_2)}{2\ln\alpha} \frac{\ln T}{T} + \frac{(\Delta_z + C_1)}{2m\ln\alpha} \frac{\ln T}{\sqrt{T}} + \frac{\alpha^2 M B_1 G^2}{2\ln\alpha} \frac{\ln T}{\sqrt{T}}. \tag{74}$$

Therefore, we complete the proof. $\qquad\square$

*Proof.* (**of Theorem 4.3**) In this theorem, we consider the boundary functions $\delta(t) = 1/\alpha^{t-1}$, and the stage length $S_t = \lceil S_0 \alpha^{t-1} \rceil$ with $S_0 = \sqrt{T}$, then we have the number of stages $N = \lfloor \log_\alpha((\alpha - 1)\sqrt{T} + 1) \rfloor$ and $\delta(N) = \alpha^{N-1}$. Next we estimate $\sum_\delta = \sum_{t=1}^{N} S_t \delta^{-1}(t) \geq (\alpha - 1)^2 T^{3/2} + 2(\alpha - 1)T$. Then applying these results into Lemma C.3, we have

$$\mathbb{E}[\|\nabla f(\hat{x}_T)\|^2] \leq \mathcal{O}\left( \frac{W_1^1}{T^{3/2}} + \frac{C_0}{T} + \frac{\Delta_z}{m\sqrt{T}} + \frac{C_1}{m\sqrt{T}} + \frac{M B_1 G^2}{\sqrt{T}} \right). \tag{75}$$

$\qquad\square$

# D SUPPLEMENTARY CONVERGENCE RESULTS

## D.1 CONVERGENCE OF SGDM WITH CONSTANT AND $1/\sqrt{t}$ BANDWIDTH DECAYING STEP-SIZE

We focus on a single stage that $N = 1$. In this case, we first consider the constant step-size $\eta_i^t = \eta_0/\sqrt{T}$. Recalling the result of Lemma 4.1 and letting $\eta_i^t = \eta_0/\sqrt{T}$ for each $i \geq 1$, we have

$$\frac{1}{T} \sum_{i=1}^{T} \mathbb{E}[\|\nabla f(x_i^1)\|^2] \leq \left( \frac{W_1^1}{T} + \frac{B_1 G^2 \eta_0}{\sqrt{T}} \right)$$

where $B_1 = r(1-\beta)(2-\beta) + \frac{L}{2(1-\beta)^2}$ and $r = \frac{\beta L}{2(1-\beta^2)(1-\beta)^2}$.

Next we turn to analyze the $1/\sqrt{t}$ bandwidth step-size $\eta_i^1 \in [m/\sqrt{i}, M/\sqrt{i}]$ (which is also monotonic decreasing). Recalling the result of Lemma 4.1

$$\mathbb{E}[W_{i+1}^t \mid \mathcal{F}_i^t] \leq W_i^t + A_1 \left( \frac{1}{\eta_i^t} - \frac{1}{\eta_{i-1}^t} \right) - \|\nabla f(x_i^t)\|^2 + \eta_i^t B_1 G^2 \tag{76}$$

and applying the result from $i = 1$ to $T$, we have

$$\frac{1}{T} \sum_{i=1}^{T} \mathbb{E}[\|\nabla f(x_i^1)\|^2] \leq \frac{1}{T} \left( (W_1^1 - \mathbb{E}[W_{T+1}^1]) + A_1 \left( \frac{1}{\eta_T^1} - \frac{1}{\eta_0^1} \right) + B_1 G^2 \sum_{i=1}^{T} \eta_i^1 \right).$$

Then applying the step-size $\eta_i^1 \in [m/\sqrt{i}, M/\sqrt{i}]$ gives

$$\frac{1}{T} \sum_{i=1}^{T} \mathbb{E}[\|\nabla f(x_i^1)\|^2] \leq \left( \frac{W_1^1}{T} + \frac{A_1}{m\sqrt{T}} + \frac{2M B_1 G^2}{\sqrt{T}} \right). \tag{77}$$

Thus, we can achieve an $\mathcal{O}(1/\sqrt{T})$ optimal rate for SGDM with $1/\sqrt{t}$ bandwidth step-size on nonconvex problems. When $M = m$, then the step-size reduces to $\eta_i^1 = m/\sqrt{i}$, we also provide the convergence guarantee for the commonly used $1/\sqrt{i}$ decaying step-size.

## D.2 CONVERGENCE GUARANTEES FOR CYCLICAL STEP-SIZES

In Loshchilov and Hutter (2017), the authors proposed a cosine annealing step-size

$$\eta^t = \eta_{\min}^t + \frac{1}{2} \left( \eta_{\max}^t - \eta_{\min}^t \right) (1 + \cos(T_{cur} \cdot \pi/T_t)). \tag{78}$$

where $\eta_{\min}^t$ and $\eta_{\max}^t$ are ranges of the step-size, and $T_{cur}$ accounts for how many epochs since the beginning of the current stage and $T_t$ accounts for the current stage length (epoch). At each stage $t$, the step-size is monotonically decaying within the range $\eta_{\min}^t$ and $\eta_{\max}^t$. In this paper, we propose a general bandwidth framework for step-size which can cover this situation as long as $m\delta(t) \leq \eta_{\min}^t$ and $\eta_{\max}^t \leq M\delta(t)$. If the ranges $\{\eta_{\min}^t, \eta_{\max}^t\}$ and the stage length are chosen as for example $1/\sqrt{t}$ in Theorems A.1 and A.2 or step-decay in Theorems 3.1 and 3.3, the theoretical convergence of SGD under the cosine annealing step-size is guaranteed by our analysis in Section 3. Moreover, because the cosine annealing is monotonic at each stage, so the convergence of SGD with momentum under the cosine annealing policy is also guaranteed by the analysis of Section 4 as long as the ranges $\eta_{\min}^t, \eta_{\max}^t$ are within our bands. To the best of our knowledge, Li et al. (2021) provides a convergence guarantee for cosine step-size. However, to achieve a near-optimal rate for the general smooth (non-convex) problems, the initial step-size is required to be bounded by $\mathcal{O}(1/\sqrt{T})$ which is obviously impractical when the total number of iteration $T$ is large (also discussed in related work). In our framework, the cosine step-size is allowed to start from a larger step-size and gradually decay. Besides, our results (e.g., Theorems 3.3 and 4.3) provide state-of-the-art convergence guarantees for cosine step-size which remove the $\log T$ term of Li et al. (2021).

Another interesting example is triangular cyclical step-size proposed by Smith (2017), which sets minimum $\eta_{\min}^t$ and maximum $\eta_{\max}^t$ boundaries and the learning rate cyclically varies (linearly

increasing then linearly decreasing) in these bounds. In each stage, the step-size is non-monotonic. In their paper, the author also consider a variant which cuts $\eta^t_{\min}$ and $\eta^t_{\max}$ in half after each stage. This is exactly the step-decay boundary we discussed. Our analysis in Section 3 can provide the convergence guarantees for such kinds of step-sizes. However, such cyclical step-size is not monotonic in each stage, so our analysis for SGDM in Section 4 is not suitable for this situation.

# E    ADDITIONAL DETAILS OF THE EXPERIMENTS ON BANDWIDTH STEP-SIZES

In this section, we provide additional details about the numerical experiments in Section 5.

## E.1    HOW TO DESIGN THE BANDWIDTH STEP-SIZES AND SELECT PARAMETERS

To better understand the bandwidth step-sizes tested in the numerical experiments, we visualize the step-size $\eta^t_i$ (y-axis is $\log(\eta^t_i)$) vs the number of epochs in Figure 5. We first consider the popular "step-decay" policy as the baseline. During the first stage, the bandwidth step-size follows the lower bound. From the second stage and on, we let the initial step-size in each stage to be equal to the upper bound and the last step-size in the stage to reach the lower bound. Our numerical experience has shown that the best performance is obtained when the initial step-size of each stage is larger than the final step-size of the previous stage, which means that the step-size experiences a sudden increase before it decreases again. For the step-decay band, we consider four decay modes: $1/i$, $1/\sqrt{i}$, linearly, and according to a cosine function (Loshchilov and Hutter, 2017) and update the step size each epoch. If the training size is $n$ and sample size per iteration is $b$, then one epoch is $n/b$ iterations.

We also adopt the polynomial $1/\sqrt{t}$ step-size as the boundary function, named $1/\sqrt{t}$-band, and update the step-size every epoch. We add similar perturbation as for the step-decay band, but we do not apply the perturbation per stage. Otherwise, the perturbation is too frequent and just increases the variance of the iterates. We tune the frequency of the perturbations (denoted as $N_0$) to undergo a similar number of cycles as the step-decay perturbations. In the first cycle, the bandwidth step-sizes agree with their lower bound, just as for step-decay. From the second cycle, we begin to add the decreasing perturbations, e.g., $1/\sqrt{i}$, $1/i$ and linearly. As discussed above, these perturbations are only adjusted between stages. Several different $1/\sqrt{t}$ bandwidth step-sizes are shown in Figure 5.

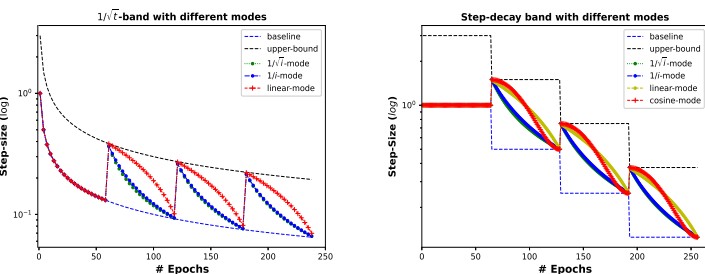

Figure 5: Bandwidth step-sizes: $1/\sqrt{t}$-band (left); step-decay-band (right)

We perform a grid search for the initial step-size $\eta_0 \in \{0.01, 0.05, 0.1, 0.5, 1, 5\}$ of the baseline step-sizes. For step-decay step-size (baseline), we select the decay factor $\alpha$ from $\{1.5, 2, 3, 4, \cdots, 12\}$ and set the number of stages $N = \lfloor \log_\alpha T/2 \rfloor$ according to Theorems 3.1 and 4.2. We choose the lower bound parameter $m = \eta_0$ to agree with the baseline. The bandwidth $s = M/m = \alpha\theta$ where $\theta \in \{0.5, 0.8, 1, 1.2, 1.3, 1.5, 1.8\}$. If $\theta > 1$, it means that the starting step-size at the current stage is larger than the ending step-size from the previous stage. For the $1/\sqrt{t}$-band step-sizes, we choose $\eta^t_i = \eta_0/(1 + a\sqrt{t})$ as the baseline , and select the best $a > 0$ to make the final step-size reach the interval $\{0.001, 0.005, 0.01, 0.05, 0.1, 0.5, 1\}$. Moreover, we select the lower bound parameter $m = \eta_0$ to make sure that the lower bound agrees with the baseline. For the upper bound parameter $M$, we do a grid search for $s = M/m \in \{2, 3, 4, 5, 6\}$. The number of perturbation cycles $N_0$ for the $1/\sqrt{t}$-band step-sizes is chosen from $\{1, 2, 3, 4\}$. All the hyper-parameters are selected to work best according to their performance on the test dataset.

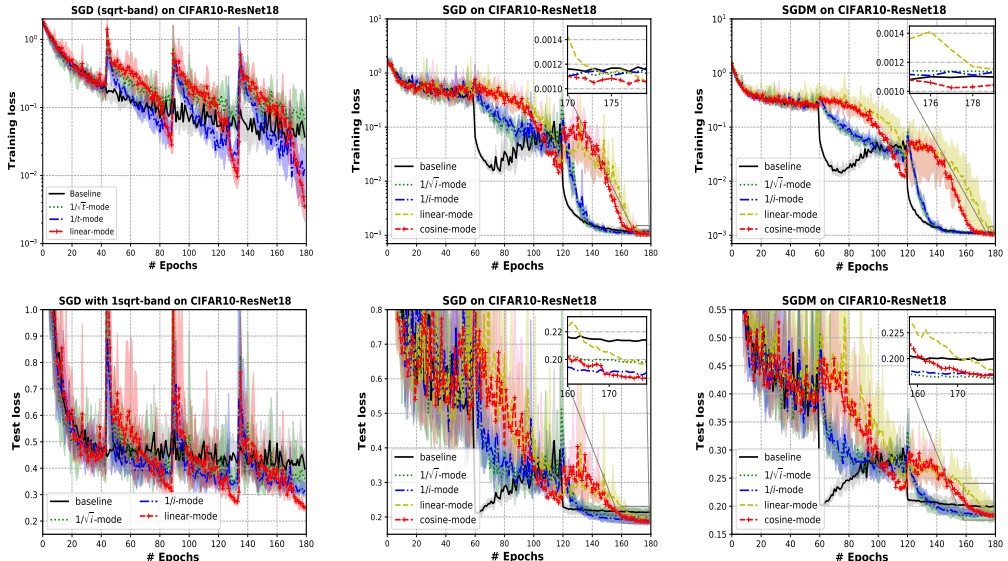

Figure 6: The training loss and test loss of $1/\sqrt{t}$-band (left column), step-decay band for SGD (middle column), and step-decay band for SGDM (right column) on CIFAR10

## E.2 EXPERIMENTS DETAILS ON CIFAR10 AND CIFAR100 DATASETS

In this subsection, we will give the implementation details for the experiments on CIFAR10 and CIFAR100. The benchmark datasets CIFAR10 and CIFAR100 (Krizhevsky, 2009) both consist of 60000 colour images (50000 training images and the rest 10000 images for testing). The maximum epochs called for the two datasets is 180 and the batch size is 128. All the experiments on CIFAR datasets are implemented in Python 3.7.4 and run on 2 x Nvidia Tesla V100 SXM2 GPUs with 32GB RAM. All experiments are repeated 5 times to eliminate the effect of the randomness. The performance of different algorithms is evaluated in terms of their loss function value and classification accuracy on the training and test datasets. All the results for training loss, test loss, and test accuracy are reported in Figures 6, 7, and 3.

For CIFAR 10, we train an 18-layer Resident Network model (He et al., 2016) called ResNet-18. We first test the vanlia SGD with a weight decay of 0.0005. The initial step-size $\eta_0 = 1$ and $a = 1.41618$ for $1/\sqrt{t}$ step-size (baseline). For $1/\sqrt{t}$-band step-sizes, we set: $s = 2$ and $N_0 = 3$ for $1/\sqrt{i}$ mode; $s = 3$ and $N_0 = 3$ for $1/i$ mode; and $s = 4$ and $N_0 = 3$ for linear mode. For the step-decay step-size (baseline) and also the bandwidth step-sizes, the initial step-size $\eta_0 = 0.5$ and decay factor $\alpha = 6$. For step-decay band step-sizes, the parameter $\theta$ is 1.3 for the $1/\sqrt{i}$, $1/i$ and linear modes, and is 1.2 for cosine mode. We also implement the SGD with momentum (SGDM) algorithm, with the momentum parameter of 0.9 and a weight decay of 0.0005. For the step-decay step-size, the initial step-size $\eta_0$ is 0.05 and the decay factor $\alpha$ is 6. We choose the same initial step-size and decay factor for the step-decay bandwidth step-sizes. The best $\theta$ is 1.3 for the four decay modes.

In a similar way, we also detail our parameter selection for the experiments on CIFAR100. On this data set, we train a $28 \times 10$ wide residual network (WRN-28-10) (Zagoruyko and Komodakis, 2016). We first implement vanilla SGD with a weight decay of 0.0005. For the baseline of the $1/\sqrt{t}$ band, we set $\eta_0 = 0.5$ and $a = 3.65224$. For the $1/\sqrt{t}$-band step-sizes we use: $s = 2$ and $N_0 = 3$ for $1/\sqrt{i}$-mode; $s = 4$ and $N_0 = 3$ for $1/i$-mode; $s = 4$ and $N_0 = 2$ for linear-mode. For the step-decay band, we choose $\eta_0 = 0.5$ and $\alpha = 6$. The parameter $\theta$ is set to 1.3 for $1/i$ mode and 1.2 for the other modes. Then we also apply the step-decay band step-sizes on the SGD with momentum (SGDM) algorithm, where the momentum parameter is 0.9 and the weight decay parameter is 0.0005. We set the initial step-size $\eta_0 = 0.1$ and $\alpha = 6$ for the baseline and other step-decay band step-sizes; $\theta = 1.2$ for $1/\sqrt{i}$ and linear modes; and $\theta = 1.3$ for $1/i$ and cosine modes.

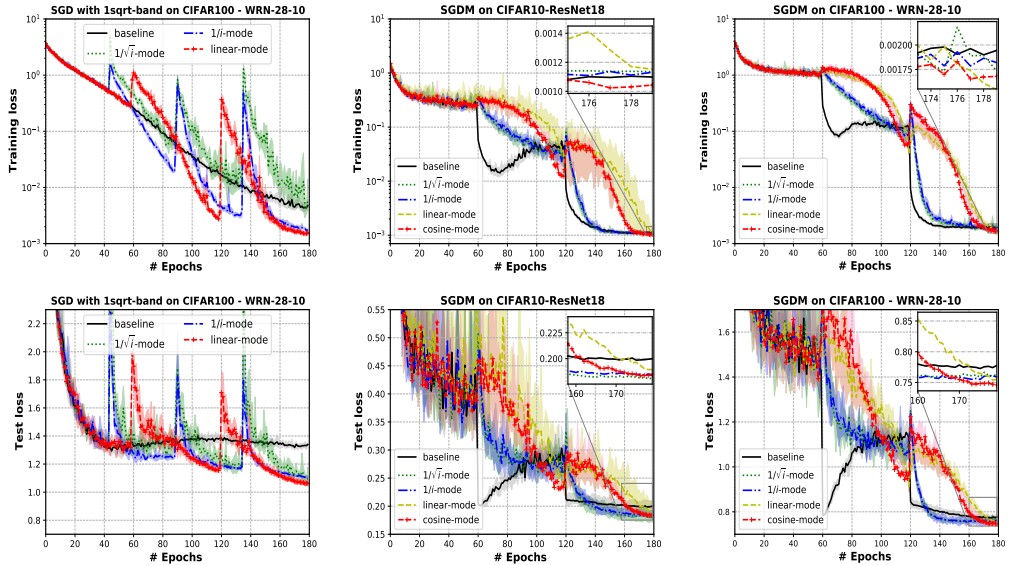

Figure 7: The training loss and test loss of $1/\sqrt{t}$-band (left column), step-decay band for SGD (middle column), and step-decay band for SGDM (right column) on CIFAR100

### E.3 EXPERIMENTS RESULTS OF THE TOY EXAMPLE ON BANDWIDTH STEP-SIZE

In this subsection, we describe the toy example and also report additional results using other bandwidth step-sizes.

The loss-function is the two-dimensional ($x, y \in \mathbb{R}$ are the variables) non-convex function:

$$f(x, y) = \left((x + 0.7)^2 + 0.1\right)(x - 0.7)^2 + (y + 0.7)^2 \left((y - 0.7)^2 + 0.1\right)$$

which has four local minima (denoted by ① to ④[3]), one of which is global (④). We execute 10000 algorithm runs with an initial point $(-0.9, 0.9)$. The total number of iterations is set to $T = 3000$. The gradient noise is drawn from the standard normal distribution. The setting of the experiments follows (Shi et al., 2020). The step-sizes we tested in this part are similar to Section E.1. The difference is that here we update the step-size per iterate instead of per epoch, as we did on the CIFAR datasets. We report the percentage (%) of the final iterate close to each local minima in Table 2. Note that the results for constant (large and small) step-size, step-decay (baseline), and step-decay with linear have already been presented in the introduction of the main document. The large constant step-size is $0.1$ and the small constant step-size is $0.05$. As we can see, $1/\sqrt{t}$-band with $1/\sqrt{i}, 1/i$ and linear modes more likely to escape the bad local minima and find the global solution than their baseline. Except the result of step-decay with linear (shown in Figure 2 and Table 1), we also find that other bandwidth step-sizes achieve good performance and work better than the baseline.

In the toy experiment, we also test the performance of step-decay (baseline) and step-decay with linear on different initial points. We evenly select 100 initial points $x_0$ from the region $[-1, 1] \times [-1, 1]$. At each initial point, we repeat the same process as previous initial point $(-0.9, 0.9)$ and do 10000 runs. We record the percentage of final iterate close to global minima ④ at each initial point in Figure 8. The $x$-axis denotes the distance of the initial point $x_0$ and the global minima $x^*$. We can see that compared to its baseline, step-decay with linear is less sensitive to the selection of the initial point and a relatively large $\theta > 1$ works better if the initial point is far from the global minima.

---

[3]Notation: ① denotes the local minima at $(-0.7, 0.7)$; ② denotes the local minima at $(0.7, 0.7)$ ; ③ denotes the local minima at $(-0.7, -0.7)$; ④ denotes the global minima at $(0.7, -0.7)$.

Table 2: The percentage (%) of the final iterate close to each local minima

| step-size | type | ① | ② | ③ | ④ |
|---|---|---|---|---|---|
| const | small | **29.61** | 24.66 | 25.13 | 20.60 |
| | large | 0.12 | 3.45 | 3.28 | **93.15** |
| $1/\sqrt{t}$-band | baseline | **54.93** | 18.65 | 19.48 | 6.94 |
| | $1/\sqrt{i}$-mode | 10.92 | 22.25 | 23.51 | **42.96** |
| | $1/i$-mode | 6.92 | 21.57 | 22.63 | **45.74** |
| | linear-mode | 3.75 | 15.86 | 16.37 | **63.97** |
| step-decay-band | baseline | 0.40 | 6.89 | 7.55 | **85.16** |
| | $1/\sqrt{i}$-mode | 0.09 | 3.73 | 4.16 | **92.02** |
| | $1/i$-mode | 0.09 | 3.55 | 3.85 | **92.51** |
| | linear-mode | 0.14 | 2.95 | 2.99 | **93.92** |
| | cosine-mode | 0.18 | 3.36 | 3.48 | **92.98** |

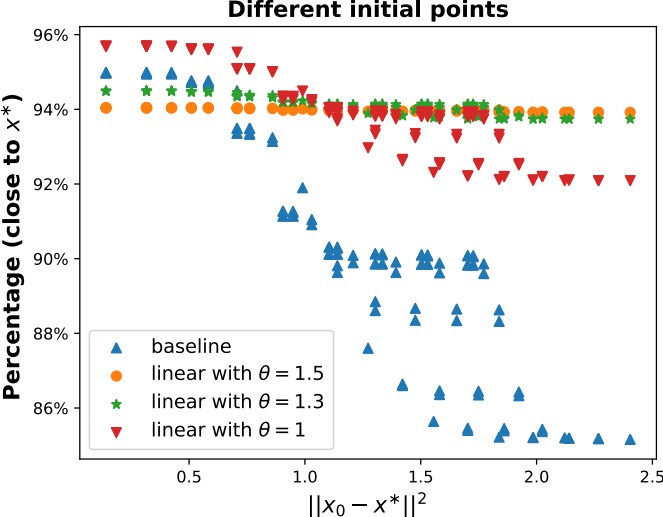

Figure 8: The toy example with different initial points for step-decay (baseline) and step-decay with linear

