# OpenReview forum: "Bandwidth-based Step-Sizes for Non-Convex  Stochastic Optimization"
_ICLR.cc/2022/Conference — ICLR 2022 Submitted_

### Official Review · Reviewer_p6jS · 2021-11-01

**Correctness:** 3
**Technical Novelty And Significance:** 3
**Empirical Novelty And Significance:** 3
**Recommendation:** 6
**Confidence:** 3

**Main Review:**

The problem and assumption setup is proper, the convergence derivation is clear and covers multiple variants of bandwidth step size application.

I have a few concerns:
regarding the theory,

1. the experiments shows that the multistage gradient methods helps avoid the local minima, can you provide any theoretical inspiration for this?

regarding to the experiments

2. why the accuracies saw a sudden increase after 60 epochs for baseline SGD and SGDM in Figure 3?

3. In the toy example, it is difficult for small constant step size to jump out of bad local minima, and large constant step size achieves comparable performance with step-decay methods. Therefore, in the numerical examples for CIFAR, why do you choose the lower bound step size $m\delta(t)$ as baseline instead of  $M\delta(t)$. Can you add the experiments for $M\delta(t)$ to show band-width methods indeed helps exploration even compared with larger step size?

4. can you design numerical experiment to illustrate the convergence rate from previous theorems?

**Summary Of The Paper:**

This paper studies the bandwidth based step size optimization mainly from theoretical guarantee and experimental validation is also provided. For step-decay SGD, the paper proves the non-asymptotic convergence and an optimal rate is derived. In the SGDM scenario, an optimal convergence result is provided and beats the result from previous literature. Bandwidth schedules are first tested on a toy example showing its power to avoid local minima, then further experiments of SGDM and SGD are performed on CIFAR dataset and achieve a better final accuracy compared with baseline.

**Summary Of The Review:**

The overall paper is well orgainzed and easy to follow. The theoretical results improves the current convergence rate than previous work. The experiments show the performance of bandwidth SGD. The connection between theorems and experiments could be stronger. Suggest marginally accept.

---

### Official Review · Reviewer_wt4Y · 2021-11-01

**Correctness:** 3
**Technical Novelty And Significance:** 3
**Empirical Novelty And Significance:** 2
**Recommendation:** 5
**Confidence:** 4

**Main Review:**

**Strengths**

The optimal rates for non-convex stochastic optimization in this paper are novel and significant for the bandwidth step size setting, especially for the case of stage-wise step decay. In particular, the SGD with momentum result with stage-wise step decay is very close to what practitioners use, modulo the fact that the result only holds for a sampled iterate rather than the last iterate, and the use of unbiased gradients which do not apply to the epoch-wise random reshuffled training. Nonetheless, these are known challenges in stochastic non-convex optimization that can be dealt separately, and the optimal rates given in this paper are still good to have. For SGD (without momentum), the fixed stage-length size tradeoff in Theorem A.1 is interesting, well-interpreted, and complemented by Theorem A.2 which is very clean in terms of its constant dependence. Although the bandwidth step size framework has already been introduced in Wang et al., 2021, the results presented in this paper are either new or improving upon existing rates. Overall, the paper is well-organized and easy to follow.


**Weaknesses and concerns**
- Another popular line of non-monotonic step sizes include stochastic line search [Vaswani et al., 2019, Paquette and Scheinberg, 2020], stochastic Polyak step sizes [Loizou et al., 2020] and its derivative [Gower et al., 2021]. As far as I'm aware, these step sizes are all non-monotonic and have upper and lower bounds, which correspond to a "band" with $m$ and $M$ given by expressions involving the Lipschitz smoothness constant and other parameters. However, as these step sizes do not have a monotonically decreasing component (like $\delta(t)=1/\sqrt{t}$), we must set $\delta(t)=\delta$ to be a constant. However, to apply Lemma B.1, we need the step sizes to be de-correlated with the randomness in the current iteration (for equation 13-15 to work, which is not the case for these methods as they use the stochastic gradients to compute the step size. A discussion of whether the analysis in this paper covers these types of "bandwidth" non-monotonic step sizes would be helpful.
- I'm a bit unsure whether the step size given by Curtis et al., 2019 can indeed be applied to the analysis in this paper. In particular, equation (70) on page 25 gives the step size for different sizes of the stochastic gradient norms. This implies that $\eta^t$ is correlated with the stochastic gradient received at the particular iteration, and we have the same issue as the point above? It would be great if the authors can clarify whether I'm missing something here about the randomness dependence.
- Toy example as evidence for "bandwidth schedule helps to avoid bad local minima": I'm not sure if this toy setup is strong enough to demonstrate what it's meant to show. It is unconvincing to me why the step size alone should allow the final iterates to concentrate around the global minimum rather than local minima. It would be helpful to see what happens with different initializations (rather than just 10000 runs from a single initial point). Additionally, the theory requires sampling the iterates, but here the final iterates are taken?
- For the deep learning experiments, plots for the training loss (or the gradient norms) can be included as well. This would help demonstrate how close the training loss approximates the test loss.
- The significance of the newly introduced schedules are mainly a combination of decreasing step size and existing cosine/linear rules, and it's hard to tell whether they are significantly better.

**References mentioned**
- [Curtis et al., 2019] A stochastic trust region algorithm based on careful step normalization
- [Vaswani et al., 2019] Painless Stochastic Gradient: Interpolation, Line-Search, and Convergence Rates
- [Paquette and Scheinberg, 2020] A stochastic line search method with expected complexity analysis
- [Loizou et al., 2020] Stochastic Polyak Step-size for SGD: An Adaptive Learning Rate for Fast Convergence
- [Gower et al., 2021] Stochastic Polyak Stepsize with a Moving Target
- [Wang et al., 2021] On the Convergence of Stochastic Gradient Descent with Bandwidth-based Step Size

**Clarifications and other comments**
1. In Section 1.2 you mentioned that in this paper, you "choose a similar sampling rule as Wang et al. 2021", which is essentially sampling iterates inversely proportional to the step size used at that iteration. Does this approach imply you need to store all the iterates? My guess is not, as long as the step size sequence does not depend on other quantities generated during the course of training, we can pre-compute it and sample (with a decoupled seed) prior to training, which can be halted pre-maturely at the sampled iterate. Is the latter approach I described typically how you would employ such sampling?


**Typos and minor issues**
- Section 1.1 second bullet point: "this is the first results that provide" --> "these are the first results that provide"
- Page 4, Section 3.1
	- Should $N$ and $S_t$ have a ceiling in their respective expressions?
	- "namely Algorithm 1 with $N=(\log_\alpha T)/2 \,(\alpha>1)$ outer loops" --> "namely Algorithm 1 with $N=(\log_\alpha T)/2$ outer loops, where $(\alpha>1)$"
	- "each with a constant length of with $S_t$" --> "each with a constant stage length of $S_t$"
- Page 5: two lines above Remark 3.4: "which allows to benefit from" --> "which allows us to benefit from"
- Page 6, below equation (5): "an iterate relationship on the form" --> "an iterate relationship of the form"
- Page 7, second paragraph:
	- missing closing parenthesis after "(see (8)"
	- "upto" --> "up to"
	- "allow to" --> "allows us to"
- Page 8: Section 5.3: Reference for CIFAR10 and CIFAR100 (Kri) should be (Krizhevsky et al. 2012)?
- Page 16: between Eq 27 and 28: "Before given the proofs" --> "Before giving the proofs"  (and on page 20)


**Summary Of The Paper:**

This paper presents a general framework for analyzing SGD with a bandwidth-based step size. This step size scheme uses a monotonically decreasing boundary function along with upper and lower bound constants to cover several non-monotonic step sizes strategies, including epoch-wise step decay, cosine annealing and triangular step sizes that are commonly adopted in training deep neural networks. They provide near-optimal convergence guarantees for smooth, non-convex functions when the boundary function is $1/\sqrt{t}$, where $t$ can be thought of as the epoch counter. The analysis works by dividing the total number of iterations into stages, where the stage length can either be constant or increasing exponentially. By appropriately selecting the number of stages and the stage length, they obtained optimal rate for the stage-wise step decay scheme. Furthermore, near-optimal and optimal rates are also provided for SGD with heavy-ball momentum under the bandwidth step size assumption that covers the stage-wise step decay setting. Lastly, they proposed non-monotonic step size schedules (step-decay with linear/cosine perturbation) and compared their empirical performance on several deep learning benchmarks.

**Summary Of The Review:**

Although the bandwidth step size has been previously introduced, the theoretical results presented in this paper are interesting and novel. Additional discussions to some relevant works mentioned above should be included. The empirical contribution is somewhat weaker, but overall I am leaning towards accepting the paper.

---

### Official Review · Reviewer_xLQE · 2021-11-02

**Correctness:** 3
**Technical Novelty And Significance:** 2
**Empirical Novelty And Significance:** 2
**Recommendation:** 3
**Confidence:** 5

**Main Review:**

The strengths of this paper are:
-  The authors formalize a class of bandwidth-based step-sizes where the learning rates are allowed to vary in a region, and are specified using some stage length $S_t$ and decreasing function $\delta(t)$. Hence the step-sizes scheme has a tendency to decrease along the training process.
- They provide the convergence guarantees for SGD and SGD-M using some choices of step sizes including the step-decay and other cyclical step-sizes. Their results matches the best convergence rate in literature, under some standard assumptions (Lipschitz smooth, bounded gradients and/or variance) and an additional condition.
- They experiment these methods for neural network training tasks on the CIFAR10 and CIFAR100 datasets.

The weaknesses of this paper are:
- The main weakness is that the authors use an additional assumption that restricts the expectation of the term $[f(x_1^t) - f_*]$ for every $t\geq 1$. This assumption is not standard and is stronger than the previous ones in literature, because it basically assumes that the algorithm can not behave too bad along the training process, when $t \geq 1$. Given the fact that the step-sizes scheme is a "perturbed" version of the diminishing learning rate, I see that this perturbation is the main reason behind this additional assumption, and therefore I think it can not be removed easily.
- The bandwidth-based step-sizes in Algorithm 1 requires a lot of parameters to tune, and it takes effort to understand what kinds of learning rate scheme fall into the scope of this paper. To my surprise, $\delta(t)$ cannot be a constant. In all theorems, we need $\delta(t) = 1/ \alpha^{t-1}$ where $\alpha$ is strictly greater than 1. The stage lengths are also chosen using some particular choices, which further restricts the setting of this paper.
- For these reasons, and since the bandwidth step size framework is introduced before, the results of this paper are not quite new. Using an additional (strong) assumption, the authors design some step-sizes schemes that are related to existing practice. Hence it is not surprising that these schemes achieve the same rate of convergence as the $\mathcal{O}(1/\sqrt{T})$ learning rate in the previous work.

Other comments:
- The stage lengths and number of stages should be integers. I think the authors might want to modify the results using floor and/or ceiling notations.
- The experiment does not really tell which step-sizes scheme is dominant, and since they all require more parameters to tune, it is not surprising that they usually give better results than the constant step-sizes. And since the theoretical results does not address the generalization performance of our algorithms, it might be more reasonable to report the training loss in the classification examples.

Some references might be added:
- Stochastic Polyak Step-size for SGD: An Adaptive Learning Rate for Fast Convergence - [Loizou et al., 2020].
- A Unified Convergence Analysis for Shuffling-Type Gradient Methods - [Nguyen et al, 2020].


**Summary Of The Paper:**

This paper investigates a general class of step-sizes for stochastic gradient algorithms. The step-sizes are chosen based on some boundary function $\delta(t)$ where $\delta(t)$ decays exponentially fast throughout the stages. Then the step-sizes are bounded between two bandwidth $m \delta(t)$ and $M \delta(t)$ where $m < M$. The bandwidth-based scheme allows some flexibilities in the choice of learning rates, although the theoretical results are restricted to some settings where every hyperparameter is carefully chosen. The authors provide the theoretical results for these settings for stochastic gradient descent (SGD) and for momentum version (SGD-M), and show that these schemes attain the standard rate of convergence for this nonconvex stochastic problem. They show the numerical results for these step-sizes using a toy example and classification example for CIFAR dataset.


**Summary Of The Review:**

The authors make some improvements over the existing literature for SGD and SGD-M algorithms, using a class of bandwidth-based step-sizes that is specified by stage lengths. The main weakness of this paper is the bounded assumption on the expected output of the problem.

Since this assumption is quite strong, it is not quite comparable with the existing works for SGD without this condition. The novel aspect of this paper is also moderate. For these reasons, I recommend rejection.

---

### Official Review · Reviewer_ij2i · 2021-11-03

**Correctness:** 3
**Technical Novelty And Significance:** 2
**Empirical Novelty And Significance:** 2
**Recommendation:** 5
**Confidence:** 3

**Main Review:**

The theoretical analysis, particularly, the analysis for momentum SGD seems to be novel. I have not looked into the literature but non-monotone learning rate seems to be not well-understood previously.


Condition (1) is the basic setup of the stepsize policy. While it is related to many cyclic stepsizes, this assumption appears to be too general,  and is not exactly equivalent to the cyclic schedule such as (Loschilov and Hutter 2017, Smith 2017). Can the author elaborate any possible difference?


The author argues (in conclusion) that "our results provide theoretical guarantees for several popular cyclic stepsizes (Loshchilov, Smith) as long as **they are tuned to lie with in our bands?**" Can the author explain this more clearly?

In the remark after Lemma 4.1 the author claims the paper uses $\beta$ irrelevant to the stepsize, and highlight that this is different from Mai and Johnason 2020, Liu et al 2020. However, this is wrong. haven't Mai and Liu both use arbitrary beta in [0,1)?

In the toy example in 5.2, do you need to consider cyclic stepsize alternate between "large" and "small"?

In the experiments, what is m and M? Is this choice consistent with those in the literature?

Since the paper discuss cyclic stepsize, can you also compare with those standard cyclic stepsize in the literature, not the stagewise stepsize?



**Summary Of The Paper:**

The paper proposes a unified framework to analyze convergence conclusion of SGD/SGDM with bandwidth step-size including popular "constant and then drop" step-size,  cosine step-size and the triangular step-size. Experiments justify the advantage of the proposed bandwidth-based decaying stepsize over standard decaying stepsize.

**Summary Of The Review:**

It occurs to me that the main contribution is some new convergence analysis of SGD(M) with non-monotone stepsize in a multistage setting. The convergence rate of SGDM improves the state-of-the-art result of momentum methods. I could be wrong, however, whether the perturbation is cosine, triangular or any cyclic rule does not really matter, which makes this work less precise in characterizing the real effectiveness of each specific cyclic rule.
Since the stepsize is decreasing, the influence of bandwidth is also decreasing, which makes the dynamic of this algorithm quite different from standard bandwidth algorithm, and more close to standard SGD. Although I agree with the author on the technical novelty in handling the non-monotoniticity, it was somewhat misleading given the title only mention Bandwidth stepsize. For these reasons, I think the paper still needs some restructure and improvement.

---

### Decision · Program_Chairs · 2022-01-20

**Decision:**

Reject

**Comment:**

The reviewers have the following remain concerns:
1. The bounded function value assumption is strong. Note that the previous works for SGD and SGD-M for other LR schemes do not necessarily need this assumption, hence it may be unfair to compare with existing results and say that this work has improvements for non-monotonic schemes. The authors also agree that it is not easy to prove and remove this assumption.
2. The novelty is limited, and the contributions are somewhat incremental. The bandwidth step size scheme was already introduced in a previous work with a very similar setting. The convergence rate for the proposed LR scheme is the same as previous works for other schemes (or only better by a logarithmic term), which makes the results incremental.
3. Some of the claims are not well supported. For example, the reviewers comment that it is not clear how the proposed bandwidth step size can help to escape local minima. Although the authors aim to show this empirically, the toy setting is not strong enough to conclude the superior performance of the proposed scheme.

We encourage the authors to improve their paper and resubmit to another venue. Here are the related suggestions:
1. The authors might try to investigate and provide a rigorous proof of how the non-monotonic step size can help to escape local minima. It also helps to characterize the effectiveness of each cyclic rule (cosine/ triangular or any other) and make clear what property (cosine/linear rules or bandwidth or non-monotonicity) contributes most in the good performance of a LR scheme.
2. It is better if the assumption on the bounded function value can be removed. In addition, a theoretical/empirical analysis on the generalization performance of the proposed scheme might also be helpful.